# Solving Truly Massive Budgeted Monotonic POMDPs with Oracle-Guided Meta-Reinforcement Learning

**Manav Vora**                                                                                          *mkvora2@illinois.edu*
*Department of Aerospace Engineering*
*University of Illinois Urbana-Champaign*
*Urbana, IL 61801, USA*

**Jonas Liang**                                                                                          *junhang2@illinois.edu*
*Department of Mathematics*
*University of Illinois Urbana-Champaign*
*Urbana, IL 61801, USA*

**Michael N. Grussing**                                                        *Michael.N.Grussing@erdc.dren.mil*
*Engineer Research and Development Center*
*U.S. Army Corps of Engineers*
*Champaign, IL 61822, USA*

**Melkior Ornik**                                                                                          *mornik@illinois.edu*
*Department of Aerospace Engineering*
*University of Illinois Urbana-Champaign*
*Urbana, IL 61801, USA*

**Reviewed on OpenReview:** *https://openreview.net/forum?id=yEAnjlmliL*

## Abstract

Many real-world decision problems, ranging from asset-maintenance scheduling to portfolio rebalancing, can be naturally modelled as budget-constrained multi-component monotonic Partially Observable Markov Decision Processes (POMDPs): each component's latent state degrades stochastically until an expensive restorative action is taken, while all assets share a fixed intervention budget. For a large numbers of assets, deriving an optimal policy for this joint POMDP is computationally intractable. To tackle this challenge, we prove that the value function of the associated belief-MDP is *budget-concave*, which allows an efficient two-step approach to finding a near-optimal policy. First, we approximate the optimal cross-component budget split via a random-forest surrogate of each single-component value function. Second, we solve each resulting budget-constrained single-component POMDP with an oracle-guided meta-trained Proximal Policy Optimization (PPO) policy: value-iteration on the fully observable counterpart yields an oracle that shapes the PPO update and greatly accelerates learning. We validate our method through experiments in two disparate domains: (i) preventive maintenance for a large-scale building infrastructure containing 1,000 components, and (ii) portfolio risk management under debit-only loss-budget constraints, where each asset's latent budget depletes with market losses and can only be replenished through costly recapitalization. Results show that our method consistently achieves longer component survival times and enhanced portfolio viability than both baseline heuristics and vanilla PPO. Furthermore, our approach maintains linear scalability in solution time with respect to the number of components.

**Code:** https://github.com/leadcatlab/Oracle-Guided-Meta-PPO

# 1 Introduction

Partially Observable Markov Decision Processes (POMDPs) offer a principled framework for sequential decision making under uncertainty regarding the true state of the system (Cassandra, 1998; Bravo et al., 2019). Solving POMDPs is computationally challenging, leading to the development of various solvers, including Monte-Carlo tree search (Katt et al., 2017), reinforcement-learning variants (Singh et al., 2021), and diverse approximation schemes (Kearns et al., 1999).

Many application domains share a *monotonic* structure, where the latent state of individual components degrades stochastically over time unless a costly restorative action is taken. Canonical examples include online advertising (Boutilier & Lu, 2016), inventory replenishment (Shin & Lee, 2015), and sequential repair or maintenance scheduling for physical assets (Miehling & Teneketzis, 2020; Bhattacharya et al., 2021). In this paper, we use monotonic POMDP to refer to this deterioration-restoration structure: without intervention, the latent condition evolves in a one-directional degrading sense, while costly maintenance actions restore or partially restore the condition. This modeling choice is aligned with structural POMDP work on partially ordered states, beliefs, and monotone policies (Lovejoy, 1987; Krishnamurthy, 2016; Miehling & Teneketzis, 2020), as well as inspection and maintenance POMDPs for deteriorating assets (Grosfeld-Nir, 2007; Papakonstantinou & Shinozuka, 2014; van Oosterom et al., 2017; Morato et al., 2022). Figure 1

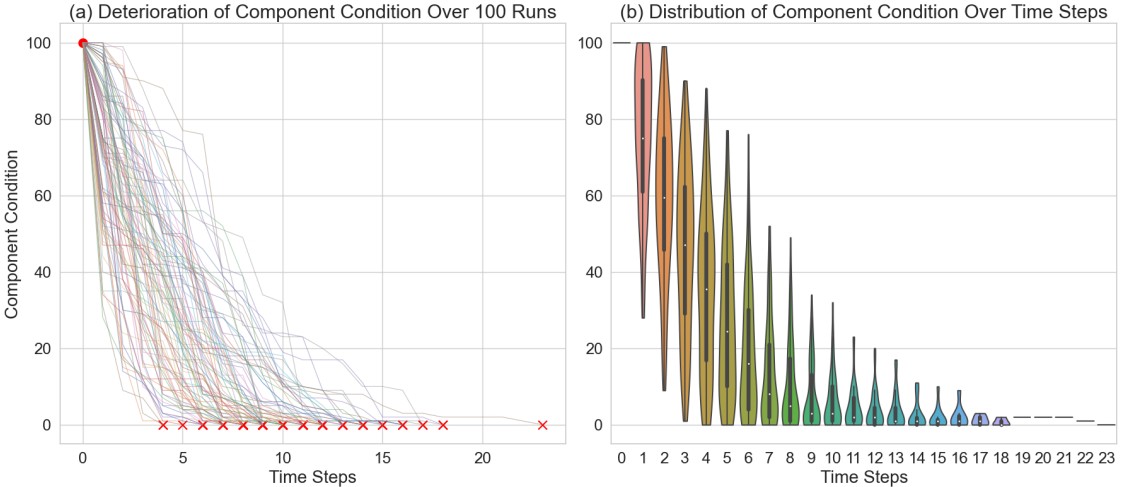

Figure 1: Condition of infrastructure component over time. (a) Line plot showing component condition over time for 100 runs. The red x marks denote the time step when condition reaches 0. (b) Violin-plot showing distribution of component condition for different time steps.

shows this stochastic decline and the probability distribution of a sample component's condition at successive time steps. While prior work, such as (Bhattacharya et al., 2020), has addressed optimal policies for single-component systems, real-world systems—from building portfolios to exchange-traded-fund (ETF) baskets—naturally involve *many* such components (Daulat et al., 2024).

In this paper, we address the challenge of computing approximately optimal policies for budget-constrained multi-component monotonic POMDPs. We assume that each component POMDP operates independently in terms of transition probabilities, but they are collectively constrained by the shared budget. Substantial work has been done to solve budget-constrained POMDPs (Lee et al., 2018; Undurti & How, 2010; Khonji et al., 2019). However, the complexity of these algorithms is often exponential in the number of states of a single POMDP. For a multi-component POMDP, where the overall state space is the Cartesian product of individual component state spaces, this complexity consequently becomes exponential in the number of components. Thus, these methods become computationally intractable for multi-component POMDPs with a large number of components. A key challenge in solving budget-constrained multi-component POMDPs is how to optimally allocate the shared budget across the multiple components. In Vora et al. (2023), the authors propose a welfare-maximization method for solving budget-constrained multi-component POMDPs.

However, the method in that paper requires generating optimal policies for multiple budget values for every component POMDP to get the optimal budget allocation. Hence, it cannot be scaled to a large number of components.

**Our insight.** The primary computational bottleneck in solving budget-constrained multi-component POMDPs is the *coupling* induced by the shared budget. If that budget could be split *a-priori*, the joint POMDP would factor into $n$ independent, single-component problems solvable in parallel. To enable this decomposition, we prove that the optimal value function of a *single* monotonic POMDP is **concave** in its allocated budget. This budget-concavity lets us decouple first, optimise second:

(1) *Budget allocation.* We maximize a concave surrogate of the value function, estimated with a random-forest regressor, to distribute the global budget across components; and

(2) *Component policies.* With budgets fixed, we learn a near-optimal policy for each component–budget pair using an *oracle-guided, meta-trained* Proximal Policy Optimization (PPO) agent, where the oracle is obtained by value iteration on the fully observable counterpart.

The result is a scalable solution whose runtime grows linearly with the number of components while retaining strong performance guarantees.

**Contributions.**

1. **Theory.** We prove budget-concavity of the optimal value function for monotonic POMDPs. While prior works implicitly assume and use this budget-concavity, our work provides the first general structural guarantee that formally links budget availability to expected return.

2. **Algorithms.** We introduce (i) a random-forest budget-allocation module that exploits concavity for fast global optimization, and (ii) an oracle-guided meta-PPO solver for each single-component POMDP.

3. **Empirical evidence.** On two domains—preventive maintenance of a 1000-component building and portfolio loss-budget management with recapitalization—we outperform baseline heuristics and vanilla PPO, whilst solution time of the proposed approach scales *linearly* in the number of components.

4. **Complexity analysis.** We provide a detailed runtime study confirming linear growth in wall-clock time as components increase from $n = 10$ to $n = 1000$.

The remainder of the paper is organized as follows. Section 2 surveys related work on budget-constrained POMDPs and large-scale maintenance or portfolio problems. Section 3 formalises the budget-constrained multi-component monotonic POMDP. Section 4 presents our solution pipeline: **(i)** Subsection 4.1 proves budget–concavity of the single-component value function; **(ii)** Subsection 4.2 exploits this structure to allocate the global budget via a random-forest surrogate; and **(iii)** Subsection 4.3 derives an oracle-guided meta-PPO policy for each component and composes them into the overall controller. Section 5 reports empirical results on infrastructure maintenance and financial loss-budget management, and Section 6 concludes with key findings and future directions.

## 2 Preliminaries and Related Work

### 2.1 Partially Observable Markov Decision Processes

A discrete-time finite-horizon Partially Observable Markov Decision Process (POMDP) (Cassandra et al., 1994) $M$ is defined by the 7-tuple $(\mathcal{S}, A, T, \Omega, O, R, H)$, which denotes the state space, action space, state transition function, observation space, observation function, reward function and planning horizon, respectively. In a POMDP, the agent does not have direct access to the true state of the environment. Instead,

the agent may maintain a *belief state*, representing a probability distribution over $\mathcal{S}$. This belief is updated based on the received observation using Bayes' rule (Araya et al., 2010).

## 2.2 POMDP Solution Methods

Computing optimal policies for a POMDP is generally PSPACE-complete (Mundhenk et al., 2000; Vlassis et al., 2012). Thus, to address the computational intractability of solving POMDPs, various approximation methods have been widely used (Poupart & Boutilier, 2002; Pineau et al., 2003; Roy et al., 2005). Several reinforcement learning approaches have also been developed for computing approximate POMDP solutions (Azizzadenesheli et al., 2016; Igl et al., 2018). However, these methods become computationally intractable when faced with the high dimensionality and shared resource constraints of budget-constrained multi-component monotonic POMDPs such as those considered in this paper.

## 2.3 Consumption MDPs and Budgeted POMDPs

The integration of budget or resource constraints into Markov Decision Processes (MDPs) has been previously studied under the frameworks of Consumption MDPs (Blahoudek et al., 2020) and Budgeted POMDPs (Vora et al., 2023). However, the algorithm proposed in Blahoudek et al. (2020) assumes full observability of the state and hence cannot be applied to budget-constrained POMDPs. A solution for budget-constrained multi-component POMDPs is presented in Vora et al. (2023). However, the method in this paper requires repeated computations of optimal policies for different budget values for all component POMDPs and hence is not scalable to a budget-constrained multi-component POMDP with a large number of components.

**Monotonic deterioration models.** In this paper, the term *monotonic POMDP* refers to the deterioration-restoration structure in our model: condition states deteriorate stochastically over time in the absence of restorative action, inspections reveal information about the latent condition, and repairs restore the condition at a cost (Bhattacharya et al., 2020; Miehling & Teneketzis, 2020; Wordsworth, 2001). Such structure appears naturally in maintenance and asset-management settings. Our contribution is to show how this structure, when present, can be exploited to scale budgeted multi-component POMDPs through a soft-budget concavity result and a componentwise decomposition.

## 3 Problem Formulation

In this paper, we consider a weakly-coupled multi-component monotonic POMDP with a total budget. Concretely, one may think of each component as an infrastructure asset whose latent condition deteriorates under continued operation and can be improved through costly maintenance, or as a financial asset whose latent risk state evolves over time and can be improved through costly recapitalization. The formal model below abstracts this common structure.

A weakly-coupled multi-component POMDP refers to a system where the individual component POMDPs have independent transition probabilities but are interconnected through a shared budget $B$. This shared budget introduces a weak coupling between the components, as the allocation of budget to one component affects the available budget for the others. The state space for an $n$-component monotonic budget-constrained POMDP is given by $\mathcal{S} = \prod_{i=1}^{n} \mathcal{S}_i$, where $\mathcal{S}_i$ represents the state space for component $i$, and $i \in \{1, \ldots, n\}$. The action space is given by $\mathcal{A} = \prod_{i=1}^{n} \mathcal{A}_i$, where the action space for component $i$ is $\mathcal{A}_i = \{d^i, q^i, m^i\}$. Here, $d^i$ denotes deferring intervention on component $i$, $q^i$ denotes inspecting component $i$, and $m^i$ denotes maintaining or repairing component $i$. Each action incurs a fixed cost. The state at time instant $k$ is an $n$-tuple, $s_k = (s_k^1, s_k^2, \cdots, s_k^n)$, where $s_k^i \in \mathcal{S}_i = \{0, 1, \ldots, \bar{s}\}$ denotes the state of component $i$, and $\bar{s} \in \mathbb{N}_0$ is the maximum possible value of $s_k^i$. Here, $\mathbb{N}_0$ denotes the set of non-negative integers. Similarly, the action at time $k$ is given by $a_k = (a_k^1, a_k^2, \cdots, a_k^n)$ and the cost associated with this action is given by $c_{a_k} = \sum_{i=1}^{n} c_{a_k^i}$, where $c_{a_k^i}$ represents the cost associated with each action $a_k^i$. The transition function for the multi-component POMDP is:

$$T(s_k, a_k, s_{k+1}) = \prod_{i=1}^{n} T^i(s_k^i, a_k^i, s_{k+1}^i).$$

The transition probability function for each component $i$ is:

$$T^i(s_k^i, a_k^i, s_{k+1}^i) = \begin{cases} p_1^i(s_k^i, a_k^i, s_{k+1}^i), & \text{if } a_k^i = m^i \text{ and} \\ & s_k^i \leq s_{k+1}^i \leq \bar{s}, \\ p_2^i(s_k^i, a_k^i, s_{k+1}^i), & \text{if } a_k^i \in \{d^i, q^i\} \\ & \text{and } s_{k+1}^i \leq s_k^i, \\ 1, & \text{if } a_k^i \in \mathcal{A}_i \text{ and} \\ & s_{k+1}^i = 0 = s_k^i, \\ 0, & \text{otherwise.} \end{cases} \tag{1}$$

Here, $p_1^i$ and $p_2^i$ are component-specific probability mass functions over the supports specified in (1). The kernel $p_1^i$ represents the stochastic repair or replacement effect of action $m^i$ and is supported only on states $s_{k+1}^i \geq s_k^i$. The kernel $p_2^i$ represents stochastic deterioration under actions $d^i$ and $q^i$ and is supported only on states $s_{k+1}^i \leq s_k^i$. Thus, action $m^i$ is restorative, with the resulting state upper bounded by $\bar{s}$; action $d^i$ corresponds to passive operation without intervention; and action $q^i$ corresponds to inspection, which reveals information about the true state of the component but does not restore the component. Moreover, $s_k^i = 0$ is an absorbing failed state for all $k, i$. The structure of (1) abstracts a common transition pattern in condition-based maintenance and asset-management models: passive operation follows a stochastic deterioration kernel over ordered condition states, while intervention changes the transition law to a repair or replacement-effect kernel. Such partially observable multi-state maintenance models are used for inspection and maintenance planning when the true condition is uncertain (Guo & Liang, 2022; Deep et al., 2023; Liu et al., 2023). Closely related Markov transition models are also standard in infrastructure asset management, where deterioration kernels model natural degradation and repair-effect kernels model stochastic improvement after maintenance or rehabilitation (Mizutani & Yuan, 2023; Abera et al., 2025). Finally, the observation probability function for each component follows the model from Vora et al. (2023), where action $q^i$ gives true state information and the other two actions provide no information about the true state.

## 3.1 Problem Statement

The primary objective of this paper is to determine a policy $\pi^*$ for this multi-component monotonic POMDP over a horizon $H$ that maximizes the total expected survival time of the components while satisfying the shared budget constraint. For component $i$, we define the horizon-truncated survival time under policy $\pi$ as

$$T_{\max}^i(\pi) := \sum_{k=0}^{H-1} \mathbf{1}\{s_k^i > 0\}.$$

Thus, $T_{\max}^i(\pi)$ counts the number of time steps during which component $i$ has not yet reached the absorbing failed state, truncated at horizon $H$. The total survival time is $T_{\max}(\pi) = \sum_{i=1}^{n} T_{\max}^i(\pi)$. Mathematically, the problem can be formulated as:

$$\max_{\pi} \left( \sum_{i=1}^{n} \mathbb{E}[T_{max}^i(\pi)] \right)$$
$$\text{s.t.} \sum_{k=0}^{H-1} c_{a_k}(\pi) \leq B. \tag{2}$$

The constraint in (2) is a hard budget constraint: the realized cumulative cost must not exceed the total budget $B$ over the horizon. Both $T_{\max}^i$ and the cumulative cost depend on the policy through the actions selected along each realized trajectory. For simplicity, we will not explicitly denote this dependence in the remainder of this paper. There are many other possible formulations of the objective of the problem statement like a *maxmin* formulation:

$$\max_{\pi} \min_{i} \mathbb{E}[T_{max}^i(\pi)]. \tag{3}$$

In this paper we consider the formulation given by (2). Section 4.2 discusses how the budget-allocation step can be modified for the max-min formulation in (3).

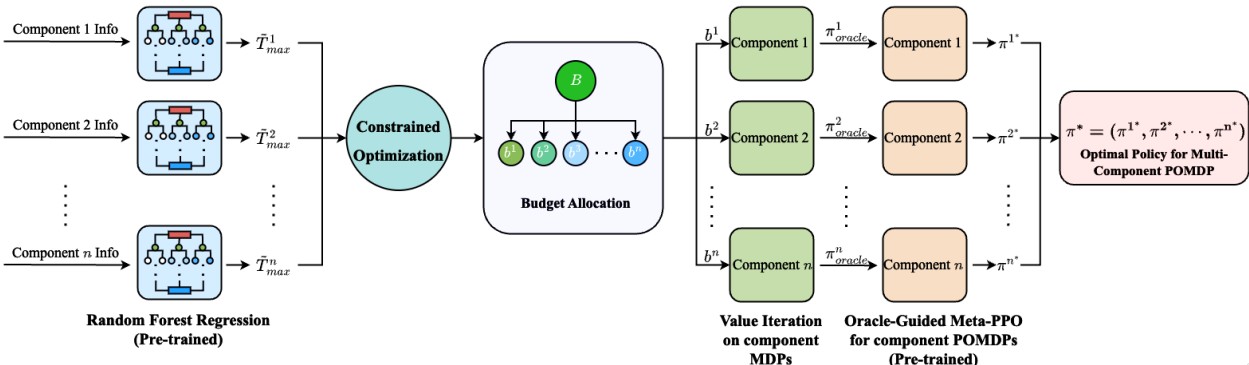

Figure 2: Architectural overview of the proposed approach.

## 4  Solution Approach

In this section, we present our methodology for solving a budget-constrained multi-component monotonic POMDP. Figure 2 presents an architectural overview of our proposed approach. The key idea is to *decouple first, optimize second.* Allocating the shared budget *as a first step of* planning shrinks the original large joint POMDP into $n$ independent single-component POMDPs. Each of these single-component POMDPs now operates with its own fixed budget cap, which is determined by the initial allocation. This transformation converts a problem that is intractable for $n \gg 1$ into $n$ modest ones that can be solved in parallel. We organize the section accordingly:

- **Structural Result: Budget Concavity** (Section 4.1): We prove that each component's value function is concave in its budget.

- **Stage 1: Budget Allocation** (Section 4.2): Leveraging concavity, we fit a random-forest surrogate of the value function and solve a tractable concave maximization problem to distribute the shared global budget across components.

- **Stage 2: Oracle-Guided Meta-PPO** (Section 4.3): With budgets fixed, we learn near-optimal policies for each component–budget pair (with respect to that component's allocated budget and local POMDP) using an oracle-guided, meta-trained PPO agent, then compose these into the overall multi-component policy.

Stage 1 distributes the joint budget once, before Stage 2 begins. A more adaptive alternative is to redistribute the residual budget periodically during execution, using the realized state and spent budget of each component. This adaptation can improve responsiveness when a component degrades or consumes budget faster than expected. However, each reallocation step must solve a new joint allocation problem because the residual budget is shared: increasing the remaining allocation to one component necessarily decreases what is available to the others. Thus, although the component transition models remain independent, the online reallocation decision couples the components through the residual budget constraint. It also requires re-estimating the per-component budget-value surrogate from the current state rather than reusing the initial-state surrogate. We evaluate this tradeoff empirically in Section 5.1.2: periodic reallocation gives only modest mean-TTF gains on the tested infrastructure subset, while the repeated surrogate refits dominate wall-clock time.

### 4.1  Budget-Concavity of the Value Function

This section clarifies the structural role of budget in a *single-component* monotonic POMDP. We distinguish between (i) a *hard* budget constraint, which requires the realized cumulative cost to never exceed the available budget, and (ii) a *soft expected-cost* constraint, which only restricts the *expected* cumulative cost. While the hard-constraint value need not be concave in general, the soft-constraint value is concave and serves as a

concave upper envelope. We further provide a quantitative bound on the soft-hard gap in terms of the budget-violation probability.

**Setting (single component)**

We consider a single-component monotonic POMDP defined by the tuple $\langle S, A, T, R, \Omega, O, \gamma \rangle$. In our problem formulation, action costs are deterministic and nonnegative; for a single component we denote the cost of taking action $a_k$ at time $k$ by $c_{a_k} \geq 0$.

Following standard POMDP practice (Cassandra, 1998), we work in the belief-MDP with belief state $b \in \Delta(S)$. Fix a horizon $H \in \mathbb{N}$ and an initial belief $b_0 = b$. For any policy $\pi$, define:

$$J_H(\pi \mid b) := \mathbb{E}^\pi \left[ \sum_{k=0}^{H-1} \rho(b_k, a_k) \;\Big|\; b_0 = b \right], \tag{4}$$

$$C_H(\pi \mid b) := \sum_{k=0}^{H-1} c_{a_k}, \qquad K_H(\pi \mid b) := \mathbb{E}^\pi [C_H(\pi \mid b)], \tag{5}$$

where $\rho(b, a) := \sum_{s \in S} b(s) R(s, a)$ is the belief reward, and the expectation is over the POMDP randomness and the internal randomization of $\pi$.

**Hard vs. soft budget feasible sets**

For a budget level $B \geq 0$, define the hard-feasible and soft-feasible policy sets:

$$\mathcal{Y}_{\text{hard}}(B) := \left\{ \pi : \; \overset{\pi}{\Pr}\big(C_H(\pi \mid b) \leq B\big) = 1 \right\}, \tag{6}$$

$$\mathcal{Y}_{\text{soft}}(B) := \{ \pi : \; K_H(\pi \mid b) \leq B \}. \tag{7}$$

The corresponding optimal values are:

$$V_H^{\text{hard}}(b, B) := \sup_{\pi \in \mathcal{Y}_{\text{hard}}(B)} J_H(\pi \mid b), \tag{8}$$

$$V_H^{\text{soft}}(b, B) := \sup_{\pi \in \mathcal{Y}_{\text{soft}}(B)} J_H(\pi \mid b). \tag{9}$$

Since $\mathcal{Y}_{\text{hard}}(B) \subseteq \mathcal{Y}_{\text{soft}}(B)$ for all $B$, we always have

$$V_H^{\text{hard}}(b, B) \;\leq\; V_H^{\text{soft}}(b, B). \tag{10}$$

**Concavity under expected-cost budgets**

We first show that the soft-budget value is concave in $B$. This is the structural property leveraged by our budget-allocation stage.

**Theorem 1** (Budget concavity under expected-cost constraint). *For any fixed belief $b \in \Delta(S)$ and horizon $H \geq 0$, the function $B \mapsto V_H^{\text{soft}}(b, B)$ is concave on $[0, \infty)$.*

*Proof.* Fix any $B_1, B_2 \geq 0$ and $\lambda \in [0, 1]$, and define $B_\lambda := \lambda B_1 + (1 - \lambda) B_2$. Let $\varepsilon > 0$. By definition of the supremum in (9), there exist policies $\pi_1, \pi_2$ such that

$$K_H(\pi_1 \mid b) \leq B_1, \qquad\qquad J_H(\pi_1 \mid b) \geq V_H^{\text{soft}}(b, B_1) - \varepsilon,$$
$$K_H(\pi_2 \mid b) \leq B_2, \qquad\qquad J_H(\pi_2 \mid b) \geq V_H^{\text{soft}}(b, B_2) - \varepsilon.$$

Construct a randomized policy $\pi_\lambda$ that draws an internal Bernoulli random variable $Z \sim \text{Bernoulli}(\lambda)$ at $k = 0$ and then executes $\pi_1$ for all $k = 0, \ldots, H-1$ if $Z = 1$ and $\pi_2$ otherwise. By the law of total expectation

and the fact that $Z$ is independent of the POMDP randomness,

$$\begin{aligned} K_H(\pi_\lambda \mid b) = \mathbb{E}[C_H(\pi_\lambda \mid b)] &= \lambda \mathbb{E}[C_H(\pi_1 \mid b)] + (1 - \lambda) \mathbb{E}[C_H(\pi_2 \mid b)] \\ &= \lambda K_H(\pi_1 \mid b) + (1 - \lambda) K_H(\pi_2 \mid b) \leq \lambda B_1 + (1 - \lambda) B_2 = B_\lambda, \end{aligned}$$

so $\pi_\lambda \in \mathcal{Y}_{\text{soft}}(B_\lambda)$. Similarly,

$$\begin{aligned} J_H(\pi_\lambda \mid b) &= \lambda J_H(\pi_1 \mid b) + (1 - \lambda) J_H(\pi_2 \mid b) \\ &\geq \lambda \big( V_H^{\text{soft}}(b, B_1) - \varepsilon \big) + (1 - \lambda) \big( V_H^{\text{soft}}(b, B_2) - \varepsilon \big). \end{aligned}$$

Since $V_H^{\text{soft}}(b, B_\lambda)$ is the supremum of $J_H(\pi \mid b)$ over all $\pi \in \mathcal{Y}_{\text{soft}}(B_\lambda)$, we have

$$V_H^{\text{soft}}(b, B_\lambda) \geq J_H(\pi_\lambda \mid b) \geq \lambda V_H^{\text{soft}}(b, B_1) + (1 - \lambda) V_H^{\text{soft}}(b, B_2) - \varepsilon.$$

Letting $\varepsilon \to 0$ yields

$$V_H^{\text{soft}}(b, B_\lambda) \geq \lambda V_H^{\text{soft}}(b, B_1) + (1 - \lambda) V_H^{\text{soft}}(b, B_2),$$

which proves concavity. $\qquad\square$

### Soft relaxation as an envelope for the hard constraint

Because $\mathcal{Y}_{\text{hard}}(B) \subseteq \mathcal{Y}_{\text{soft}}(B)$, the soft value $V_H^{\text{soft}}(b, B)$ is always an upper bound on the hard value $V_H^{\text{hard}}(b, B)$. To better understand when this upper bound is exact and when it may be loose, we examine the relationship between the two values at a fixed budget level $B \geq 0$.

For the remainder of this discussion, assume that the supremum in (9) is attained at some soft-optimal policy $\pi_{\text{soft}}^*(B) \in \mathcal{Y}_{\text{soft}}(B)$, i.e.,

$$J_H(\pi_{\text{soft}}^*(B) \mid b) = V_H^{\text{soft}}(b, B).$$

Under this local attainment assumption, we distinguish two cases depending on whether $\pi_{\text{soft}}^*(B)$ is also hard-feasible.

**Case 1:** $\pi_{\text{soft}}^*(B) \in \mathcal{Y}_{\text{hard}}(B)$. In this case the relaxation is exact: the optimal values coincide, and any soft-optimal policy that is hard-feasible is also hard-optimal.

**Proposition 2** (Membership implies tightness). *If $\pi_{\text{soft}}^*(B) \in \mathcal{Y}_{\text{hard}}(B)$, then*

$$V_H^{\text{hard}}(b, B) = V_H^{\text{soft}}(b, B),$$

*and $\pi_{\text{soft}}^*(B)$ attains the hard supremum, i.e.,*

$$J_H(\pi_{\text{soft}}^*(B) \mid b) = V_H^{\text{hard}}(b, B).$$

*Proof.* First, by $\mathcal{Y}_{\text{hard}}(B) \subseteq \mathcal{Y}_{\text{soft}}(B)$ we have $V_H^{\text{hard}}(b, B) \leq V_H^{\text{soft}}(b, B)$.

Assume $\pi_{\text{soft}}^*(B) \in \mathcal{Y}_{\text{hard}}(B)$. Then $\pi_{\text{soft}}^*(B)$ is feasible for the hard problem, so by the definition of the hard supremum,

$$V_H^{\text{hard}}(b, B) \;\geq\; J_H(\pi_{\text{soft}}^*(B) \mid b) \;=\; V_H^{\text{soft}}(b, B),$$

Together with $V_H^{\text{hard}}(b, B) \leq V_H^{\text{soft}}(b, B)$ this implies equality: $V_H^{\text{hard}}(b, B) = V_H^{\text{soft}}(b, B)$. Consequently, it follows directly that $\pi_{\text{soft}}^*(B)$ attains the hard supremum.

$\qquad\square$

**Case 2:** $\pi_{\text{soft}}^*(B) \notin \mathcal{Y}_{\text{hard}}(B)$. In this case $V_H^{\text{soft}}(b, B)$ may strictly exceed $V_H^{\text{hard}}(b, B)$; we bound the gap in terms of the budget-violation probability under $\pi_{\text{soft}}^*(B)$. The key idea is to compare a possibly budget-violating policy to the same policy stopped at its first budget-violation point.

**Budget truncation.** Fix $B \geq 0$ and a policy $\pi$. Define the remaining-budget process

$$B_0 := B, \qquad B_{k+1} := B_k - c_{a_k}, \qquad k = 0, \ldots, H-1,$$

where $a_k$ is the action selected by $\pi$ at time $k$ (under the belief-MDP, $\pi$ depends on the current belief and remaining budget). Define the first budget-violation time as

$$\tau_B^\pi := \inf \{k \in \{0, \ldots, H-1\} : c_{a_k} > B_k\},$$

with the convention $\tau_B^\pi = H$ if the set is empty. Define the truncated policy $\pi_B^{\mathrm{tr}}$ as a wrapper around $\pi$: at each time $k$, given $(b_k, B_k)$, sample $a_k$ according to $\pi$; if $c_{a_k} \leq B_k$, execute $a_k$ and update $B_{k+1} = B_k - c_{a_k}$; otherwise if $c_{a_k} > B_k$, terminate and transition to a zero-reward absorbing state. By construction, $\pi_B^{\mathrm{tr}} \in \mathcal{Y}_{\mathrm{hard}}(B)$.

**Proposition 3** (Soft-hard gap via violation probability). *Assume the per-step reward is bounded above by* 1, *i.e.,* $R(s,a) \leq 1$ *for all* $(s,a)$. *Let* $\tau_B^*$ *denote the first budget-violation time under* $\pi_{\mathrm{soft}}^*(B)$, *and let* $(\pi_{\mathrm{soft}}^*(B))_B^{\mathrm{tr}}$ *denote the corresponding budget-truncated policy. Then*

$$0 \leq J_H(\pi_{\mathrm{soft}}^*(B) \mid b) - J_H\big((\pi_{\mathrm{soft}}^*(B))_B^{\mathrm{tr}} \mid b\big) \leq H \cdot \overset{\pi_{\mathrm{soft}}^*(B)}{\Pr}(\tau_B^* < H),$$

*and consequently,*

$$0 \leq V_H^{\mathrm{soft}}(b, B) - V_H^{\mathrm{hard}}(b, B) \leq H \cdot \overset{\pi_{\mathrm{soft}}^*(B)}{\Pr}\big(C_H(\pi_{\mathrm{soft}}^*(B) \mid b) > B\big).$$

*Proof.* First, note that $(\pi_{\mathrm{soft}}^*(B))_B^{\mathrm{tr}} \in \mathcal{Y}_{\mathrm{hard}}(B) \subseteq \mathcal{Y}_{\mathrm{soft}}(B)$. Since $\pi_{\mathrm{soft}}^*(B)$ attains the soft supremum (attainment assumption), we have

$$J_H(\pi_{\mathrm{soft}}^*(B) \mid b) \geq J_H\big((\pi_{\mathrm{soft}}^*(B))_B^{\mathrm{tr}} \mid b\big),$$

which establishes the left inequality.

By construction, $\pi_{\mathrm{soft}}^*(B)$ and $(\pi_{\mathrm{soft}}^*(B))_B^{\mathrm{tr}}$ induce identical trajectories up to time $\tau_B^* - 1$, and the truncated policy collects zero reward from time $\tau_B^*$ onward. Therefore,

$$J_H\big((\pi_{\mathrm{soft}}^*(B))_B^{\mathrm{tr}} \mid b\big) = \mathbb{E}^{\pi_{\mathrm{soft}}^*(B)}\left[\sum_{k=0}^{\tau_B^*-1} R(s_k, a_k)\right],$$

where the sum is understood to be zero if $\tau_B^* = 0$. Hence,

$$J_H(\pi_{\mathrm{soft}}^*(B) \mid b) - J_H\big((\pi_{\mathrm{soft}}^*(B))_B^{\mathrm{tr}} \mid b\big) = \mathbb{E}^{\pi_{\mathrm{soft}}^*(B)}\left[\sum_{k=\tau_B^*}^{H-1} R(s_k, a_k)\right],$$

with the convention that this sum is zero when $\tau_B^* = H$. If $\tau_B^* = H$, then the sum is empty and equals 0. If $\tau_B^* < H$, then the sum contains at most $H - \tau_B^* \leq H$ terms and each term is at most 1, so

$$\sum_{k=\tau_B^*}^{H-1} R(s_k, a_k) \leq H.$$

Combining the two cases yields the pathwise bound

$$\sum_{k=\tau_B^*}^{H-1} R(s_k, a_k) \leq H \mathbf{1}\{\tau_B^* < H\}.$$

Taking expectations and using $\mathbb{E}[\mathbf{1}\{\tau_B^* < H\}] = \Pr(\tau_B^* < H)$ gives

$$J_H(\pi_{\mathrm{soft}}^*(B) \mid b) - J_H\big((\pi_{\mathrm{soft}}^*(B))_B^{\mathrm{tr}} \mid b\big) \leq H \cdot \overset{\pi_{\mathrm{soft}}^*(B)}{\Pr}(\tau_B^* < H).$$

Finally, because $c_{a_k} \geq 0$, once the cumulative cost exceeds $B$ it cannot decrease below $B$ later in the horizon. Hence,

$$\{\tau_B^* < H\} = \{\tau_B^* \in \{0, \dots, H-1\}\} = \left\{\sum_{k=0}^{H-1} c_{a_k} > B\right\} = \{C_H(\pi_{\text{soft}}^*(B) \mid b) > B\}.$$

Now, $(\pi_{\text{soft}}^*(B))_B^{\text{tr}} \in \mathcal{Y}_{\text{hard}}(B)$ by construction, so

$$V_H^{\text{hard}}(b, B) \geq J_H\big((\pi_{\text{soft}}^*(B))_B^{\text{tr}} \mid b\big).$$

Also, by the attainment assumption,

$$V_H^{\text{soft}}(b, B) = J_H(\pi_{\text{soft}}^*(B) \mid b).$$

Therefore,

$$V_H^{\text{soft}}(b, B) - V_H^{\text{hard}}(b, B) \leq J_H(\pi_{\text{soft}}^*(B) \mid b) - J_H\big((\pi_{\text{soft}}^*(B))_B^{\text{tr}} \mid b\big) \leq H \cdot \overset{\pi_{\text{soft}}^*(B)}{\text{Pr}}\big(C_H(\pi_{\text{soft}}^*(B) \mid b) > B\big),$$

as claimed. $\qquad\square$

**Remark 1** (Lagrangian view of the expected-cost relaxation). *A common way to obtain $\pi_{soft}^*$ is via the Lagrangian relaxation of the expected-cost constraint. In particular, for $\lambda \geq 0$ one considers the unconstrained objective*

$$\mathcal{L}_\lambda(\pi \mid b) := J_H(\pi \mid b) - \lambda\big(K_H(\pi \mid b) - B\big),$$

*and optimizes $\sup_\pi \mathcal{L}_\lambda(\pi \mid b)$. Under standard conditions for finite-horizon constrained POMDP formulations, there exists a multiplier $\lambda^\star$ such that any optimizer of $\mathcal{L}_{\lambda^\star}$ is optimal for the soft problem at budget $B$. In this subsection we assume there exists a soft-optimal policy $\pi_{\text{soft}}^*(B)$ such that $\pi_{\text{soft}}^*(B)$ maximizes $\mathcal{L}_{\lambda^\star}(\cdot \mid b)$.*

**Lemma 4** (Expected-cost slack implies small budget-violation probability). *Let $c_{\max} := \max_{a \in A} c_a$ (finite since $A$ is finite). For any policy $\pi$ and any $\Delta > 0$,*

$$K_H(\pi \mid b) \leq B - \Delta \quad \implies \quad \overset{\pi}{\text{Pr}}\big(C_H(\pi \mid b) > B\big) \leq \exp\left(-\frac{2\Delta^2}{Hc_{\max}^2}\right).$$

*Proof.* Define the Doob martingale (Doob, 1953) $M_k := \mathbb{E}^\pi[C_H(\pi \mid b) \mid \mathcal{F}_k]$, where $\mathcal{F}_k$ is the history up to time $k$. Then $M_0 = \mathbb{E}^\pi[C_H(\pi \mid b)] = K_H(\pi \mid b)$ and $M_H = C_H(\pi \mid b)$. Since each per-step cost lies in $[0, c_{\max}]$, the martingale differences satisfy $|M_k - M_{k-1}| \leq c_{\max}$ almost surely. Therefore, by the Azuma–Hoeffding inequality (Azuma, 1967; Hoeffding, 1963), for any $t > 0$,

$$\overset{\pi}{\text{Pr}}(M_H - M_0 \geq t) \leq \exp\left(-\frac{2t^2}{Hc_{\max}^2}\right).$$

Because $M_H = C_H(\pi \mid b)$ and $M_0 = K_H(\pi \mid b)$, this is equivalently

$$\overset{\pi}{\text{Pr}}(C_H(\pi \mid b) - K_H(\pi \mid b) \geq t) \leq \exp\left(-\frac{2t^2}{Hc_{\max}^2}\right).$$

If $K_H(\pi \mid b) \leq B - \Delta$, then $\{C_H(\pi \mid b) > B\} \subseteq \{C_H(\pi \mid b) - K_H(\pi \mid b) \geq \Delta\}$, so the claim follows by setting $t = \Delta$. $\qquad\square$

**Corollary 5** (Exponential soft–hard gap under expected-cost slack). *If the soft-optimal policy satisfies $K_H(\pi_{\text{soft}}^*(B) \mid b) \leq B - \Delta$ for some $\Delta > 0$, then*

$$0 \leq V_H^{\text{soft}}(b, B) - V_H^{\text{hard}}(b, B) \leq H \exp\left(-\frac{2\Delta^2}{Hc_{\max}^2}\right).$$

*Proof.* Combine Proposition 3 with Lemma 4 applied to $\pi = \pi_{\text{soft}}^*(B)$. $\qquad\square$

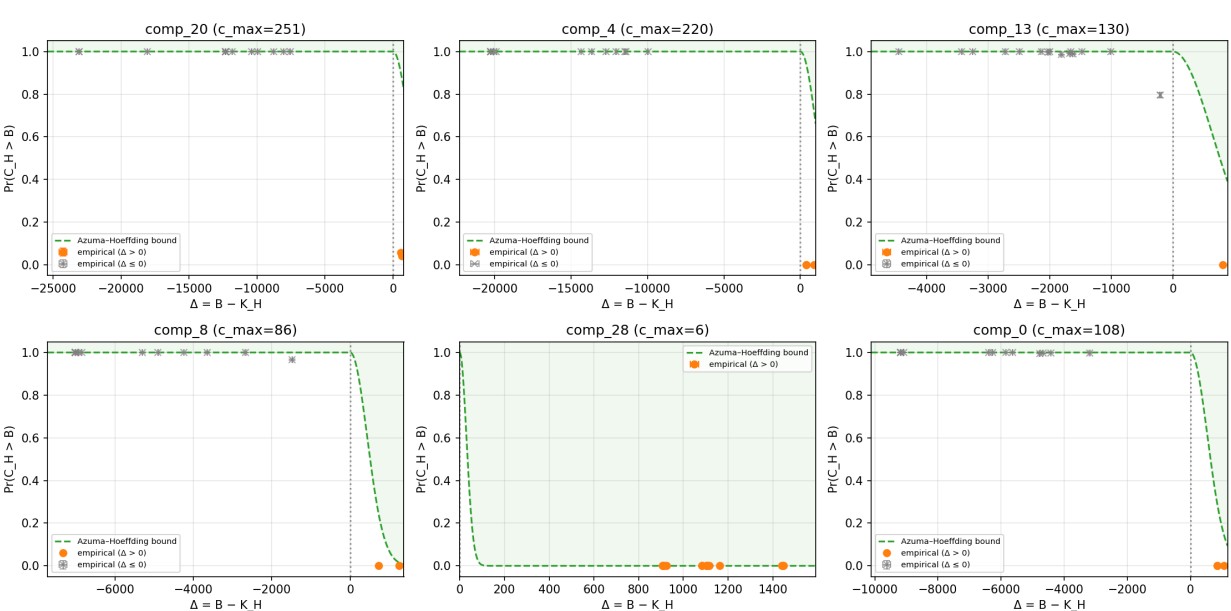

Figure 3: Empirical validation of the soft–hard gap bound. Each panel shows one infrastructure component. The horizontal axis is the expected-cost slack $\Delta = B - K_H(\pi_\lambda \mid b)$ obtained by sweeping the Lagrange multiplier $\lambda$, and the vertical axis is the empirical hard-budget violation probability estimated from rollouts. The dashed curve is the Azuma-Hoeffding upper bound $\min\{1, \exp(-2\Delta^2/(Hc_{\max}^2))\}$. Empirical violation probabilities are consistently below the theoretical curve, confirming that positive expected-cost slack yields small hard-budget violation probability in the operating regimes considered.

**Implications of Propositions 2–3.** Proposition 2 identifies when the expected-cost relaxation is exact: if a soft-optimal policy is hard-feasible, then the soft concavity result applies directly to the hard-budget objective at budget $B$. When this is not the case, Proposition 3 bounds the approximation error by a budget-violation probability term. Lemma 4 and Corollary 5 further show that whenever the soft-optimal policy leaves a nontrivial expected-cost slack, this violation probability, and hence the soft-hard gap, is exponentially small in the slack.

**Empirical validation of the violation-probability bound.** To verify that the bound above is meaningful for the component dynamics used in our experiments, we solve the Lagrangian-relaxed single-component problem over a sweep of multipliers and estimate the realized violation probability by Monte Carlo rollouts. For each of six representative real-world infrastructure components, we sweep $\lambda$ over a 16-point logarithmic grid scaled by $1/c_{\max}$, compute the corresponding Lagrangian policy, and estimate both $K_H$ and $\Pr(C_H > B)$ from $5 \times 1500$ trajectories. Figure 3 plots the empirical violation probability against $\Delta = B - K_H$, together with the Doob-martingale/Azuma-Hoeffding bound from Lemma 4. The empirical points lie below the theoretical curve in all panels. As expected for a worst-case concentration inequality, the bound is conservative for high-cost components, but it correctly captures the sharp transition from frequent violations when $\Delta < 0$ to near-zero violations once the Lagrangian policy leaves positive expected-cost slack.

### 4.1.1 Relating $\mathbb{E}[T_{\max}]$ to the Value Function

We now specialize to the survival-time objective used in our budget allocation stage.

**Lemma 6** (Expected-time equivalence)**.** *Consider the reward function*

$$R(s, a) = \begin{cases} 1, & s \neq 0, \\ 0, & s = 0, \end{cases} \tag{11}$$

*where $s = 0$ is absorbing. Let $V_H^{\text{hard}}(b, B)$ be the optimal hard-budget value in* (8) *under this reward. Denote by* $\mathbb{E}[T_{\max}(B)]$ *the expected time (up to horizon H) to reach the absorbing state under an optimal hard-budget-feasible policy. Then*

$$V_H^{\text{hard}}(b, B) = \mathbb{E}[T_{\max}(B)].$$

*Proof.* Under (11), each non-absorbing step contributes exactly 1 reward and each absorbing step contributes 0. Hence, for any hard-budget-feasible policy $\pi \in \mathcal{Y}_{\text{hard}}(B)$,

$$J_H(\pi \mid b) = \mathbb{E}^\pi \left[ \sum_{k=0}^{H-1} \mathbf{1}\{s_k \neq 0\} \right] = \mathbb{E}[T_{\max}(\pi)].$$

Maximizing both sides over $\pi \in \mathcal{Y}_{\text{hard}}(B)$ yields $V_H^{\text{hard}}(b, B) = \mathbb{E}[T_{\max}(B)]$. $\qquad\square$

**Corollary 7.** *Under the reward* (11), *the soft-budget value $V_H^{\text{soft}}(b, B)$ is a concave function of B by Theorem 1 and satisfies*

$$\mathbb{E}[T_{\max}(B)] = V_H^{\text{hard}}(b, B) \ \leq \ V_H^{\text{soft}}(b, B).$$

*Moreover, Proposition 3 gives*

$$0 \ \leq \ V_H^{\text{soft}}(b, B) - \mathbb{E}[T_{\max}(B)] \ \leq \ H \cdot \overset{\pi_{\text{soft}}^*(B)}{\Pr} \left( C_H(\pi_{\text{soft}}^*(B) \mid b) > B \right).$$

**Connection to our penalized training objective.** In Section 4.3, our PPO reward assigns a large negative penalty at every step where the *running cumulative cost* exceeds $B$, which empirically yields near-zero budget violations in rollouts. Proposition 3 and Corollary 5 together clarify why controlling budget-violation probability (or inducing expected-cost slack) is the right structural signal for tightness of the soft envelope: when violations are rare, the difference between soft and hard objectives is small.

## 4.2 Random Forest Approach for Optimal Budget Allocation

By Corollary 7, the expected maximal survival time $\mathbb{E}[T_{\max}(B)]$ is a concave function of the budget allocated to a single component. This structural property lets us treat budget splitting across $n$ components as a *concave maximization* problem—one that is both tractable and amenable to surrogate modeling. Each component evolves independently but competes for the shared budget, rendering the components weakly coupled. While reinforcement learning algorithms have made significant advances, they often face challenges when scaling to the extremely large state and action spaces characteristic of multi-component systems (Sutton & Barto, 2018). To address this scalability issue, our remedy is an *a-priori* budget distribution that decouples the system. Concretely, for component $i$ we approximate the concave map $B \mapsto \mathbb{E}[T_{\max}^i(B)]$ by the exponential surrogate

$$\widetilde{T}_{\max}^i(B) \ = \ \alpha^i \, e^{\beta^i B} + \gamma^i, \tag{12}$$

where $(\alpha^i, \beta^i, \gamma^i)$ are constants. While many other concave functions could be used to model $\widetilde{T}_{\max}^i$, we empirically observe that the exponential function provides a good fit for the data (see Appendix A). The allocation step only requires a tractable concave approximation of each component's budget-value curve. The exponential family is well-matched to this setting because the finite horizon imposes saturation at large budgets, and because the marginal value near zero budget remains finite, allowing the optimizer to assign zero or very small budgets to low-marginal-value components. In Section 5.1.2 and Appendix B, we empirically justify this choice by comparing against logarithmic, power-law, Hill/Michaelis-Menten, hyperbolic tangent, and piecewise linear concave surrogates.

We use a random forest regressor (Breiman, 2001) to estimate the parameters of this exponential function. The training dataset for this model is obtained via non-linear least squares regression on multiple $(\mathbb{E}[T_{\max}], b)$ pairs for various budget-constrained single-component monotonic POMDPs. The input to this model includes specific statistics related to the POMDP's transition function, which are the expected time to reach state 0 without repairs, $\mathbb{E}[T]$, and the variance of this expected time, $\sigma^2_{\mathbb{E}[T]}$, as well as the various actions costs.

Let $b^i$ denote the budget assigned to component $i$ and $\widetilde{T}^i_{\max}$ its surrogate survival time. For the sum objective in (2), the allocation problem becomes

$$
\begin{aligned}
\max_{b^{1:n}} \quad & \sum_{i=1}^{n} \widetilde{T}^i_{\max}(b^i) \\
\text{s.t.} \quad & \sum_{i=1}^{n} b^i \ \le \ B, \qquad b^i \ \ge \ 0 \ \ \forall i.
\end{aligned}
\tag{13}
$$

Because each surrogate in (12) is concave and the constraints are linear, (13) is a convex optimization problem in the standard concave-maximization form. We solve it using off-the-shelf convex optimizers. Solving (13) yields the approximately optimal budget allocation among the individual components for the objective in (2).

The same fitted per-component budget-value curves can also be used with other global objectives. For example, under the max-min objective in (3), one can introduce an auxiliary variable $\eta$ representing the minimum component value and solve

$$
\max_{b^{1:n},\eta} \ \eta \quad \text{s.t.} \quad \eta \le \widetilde{T}^i_{\max}(b^i) \ \forall i, \quad \sum_{i=1}^{n} b^i \le B, \quad b^i \ge 0.
$$

This formulation is also convex whenever each surrogate $\widetilde{T}^i_{\max}$ is concave: the constraint $\eta \le \widetilde{T}^i_{\max}(b^i)$ describes the hypograph of a concave function, and the remaining budget constraints are linear. Thus, changing the global objective changes the allocation criterion, but it does not change the underlying single-component budget-value modeling step. In this paper, we focus on the sum objective in (2).

The next subsection shows how an oracle-guided meta-PPO agent learns the individual component policies given this budget allocation.

## 4.3 Oracle-Guided RL for a Budget-Constrained Single Component

Given the per-component budgets $b^i$ obtained in Section 4.2, we now derive a near-optimal control policy for each single-component budget-constrained monotonic POMDP. We adopt the budget-augmented POMDP (bPOMDP) formalism of Vora et al. (2023), in which the state includes an additional, fully-observable coordinate that tracks cumulative cost.

The oracle policy is denoted as $\pi_{\text{oracle}}$ and is obtained by solving the corresponding MDP using value iteration. For a single-component monotonic POMDP with budget $B$, the corresponding MDP has an action space $\mathcal{A}_{\text{MDP}} = \{d, m\}$, identical transition probabilities as the POMDP, and *full observability of the state*. We then train a Proximal Policy optimization (PPO) agent (Schulman et al., 2017) that *queries* this oracle selectively: at each time step it chooses either to inspect ($q$) or to defer ($\neg q$), in which case the action recommended by the oracle is executed. Since the full state is not observable in a POMDP, we utilize the belief $b_s$ for planning. The agent's belief of the true state is updated at each time step using a particle filter approach. For our work, we empirically observe that using the expected belief $\bar{b}_s$ and the variance of the belief $\sigma^2_{b_s}$ suffices for planning.

Hence, for the proposed oracle policy-guided PPO agent, the state at time instant $k$ is given by the vector $[\bar{b}_{s_k}, c_k, \sigma^2_{b_{s_k}}]$. Furthermore, the reward function is defined as follows:

$$
R(s_k, c_k, a_k) = \begin{cases} r_1 < 0, & \text{if } c_k > B, \\ r_2 < 0, & \text{if } \lfloor \bar{b}_{s_k} \rfloor = 0, \\ r_3 = \frac{k}{H} - \alpha |\bar{b}_{s_k} - s_k|, & \text{if } \bar{b}_{s_k}, c_k > 0, \end{cases}
$$

where $|r_1| > |r_2| > |r_3|$ for all $k$, $0 < \alpha < 1$ and $\lfloor . \rfloor$ denotes the floor function. This reward function imposes substantial negative rewards for exceeding the budget $B$ and allowing the state $s_k$ to reach 0. Additionally, at each time step, the agent receives a positive reward proportional to the time step for maintaining $s_k$ above zero and incurs a penalty proportional to the absolute error between the expected belief and the true state.

As a result, the agent gets higher rewards for keeping $s_k > 0$ for a longer time and is heavily penalized when the expected belief deviates significantly from the true state. It is crucial to note that during training, the agent relies solely on the observed reward signals, without access to the true state.

### 4.4 Optimal Policy for Multi-Component Monotonic POMDPs

We now integrate the approaches described in Section 4.2 and Section 4.3 to compute the optimal policy for an $n$-component POMDP, where $n$ is substantially large. Utilizing the random forest regressor, we efficiently approximate $\mathbb{E}[T_{\max}]$ for each component $i$. Additionally, we meta-train the oracle-guided PPO agent by continuously updating the policy network's parameters over a randomly selected subset of components and budget values. This approach allows the agent to generalize across components. This meta-trained agent is then utilized to derive the optimal policy $\pi^{i^*}$ for each component $i$, following the optimal budget allocation obtained from (13). Consequently, the overall policy for the multi-component POMDP is:

$$\pi^*(s_k, a_k) = (\pi^{1^*}(s_k^1, a_k^1), \pi^{2^*}(s_k^2, a_k^2), \cdots, \pi^{n^*}(s_k^n, a_k^n)).$$

While this policy is not guaranteed to be globally optimal for the entire multi-component POMDP, we empirically observe that it performs well in practice while respecting the budget constraints. We validate this approach by evaluating its performance on real-world data in the subsequent section.

## 5 Implementation and Evaluation

In this section, we empirically validate our proposed framework on two disparate domains. The first domain, which we call the *infrastructure scenario*, involves preventive maintenance for a large-scale building comprising 1000 independent components whose latent condition stochastically degrades over time; our goal is to allocate a finite maintenance budget to maximize the expected survival time of all components. The second domain, the *financial loss-budget scenario*, addresses portfolio risk management using daily price data for S&P 500 constituents, where each asset is endowed with a debit-only loss budget that depletes under negative returns and can be replenished only through costly recapitalization. In the infrastructure scenario, we compare our oracle-guided meta-PPO approach against baseline heuristics, vanilla PPO, and an idealized oracle policy, reporting results on survival time, cost efficiency, and computational scalability across a range of budget levels. In the financial loss-budget scenario, we focus on analyzing the learned recapitalization policy and assessing the generalizability and window robustness of proposed oracle-guided meta-PPO.

### 5.1 Implementation and Evaluation for Infrastructure Scenario

In this section, we evaluate the efficacy of the proposed methodology for determining the optimal policy for a very large multi-component budget-constrained POMDP. Specifically, we compare our approach against existing methods in the context of a multi-component building maintenance scenario managed by a team of agents. We also perform a computational complexity analysis of the proposed approach, for varying number of components.

We consider an administrative building comprising 1000 infrastructure components, including roofing elements, water fountains, lighting systems, and boilers. Each component's health is quantified by the Condition Index (CI) (Grussing et al., 2006), which ranges from 0 to 100. For each infrastructure component, we utilize historical CI data to generate the transition probabilities for the corresponding POMDP, modeled using the Weibull distribution (Grussing et al., 2006). We use the `weibull_min` class from the `scipy.stats` module in Python to simulate the CI transitions over time. While a seed can be set using the `random_state` parameter in `weibull_min` for reproducibility, we did not set one to preserve the stochastic nature of the CI transitions. The condition index deteriorates stochastically over time, influenced by various factors, and can only be accurately assessed through explicit inspections, which incur a cost. A component is considered to have failed when its CI falls below a failure threshold, which is assumed to be 0. Components can be repaired to increase their CI. The building is allocated a maintenance budget of $B = 500{,}000$ units for a given horizon of 100 decision steps. At the beginning of the horizon, the CI of all components is 100. The objective of the agents is to maximize the time until failure of the components by efficiently allocating the budget among the

components and performing repairs and inspections as needed. The replacement costs (ranging from 50 to 500 units) and inspection costs (ranging from 1 to 5 units) of these components are derived from industry averages. Consistent with the approach described in Section 4.3, we model this objective as a POMDP (with $\alpha = 10^{-3}$ in the reward function). This POMDP has roughly $10^{2000}$ states and $3^{1000}$ actions.

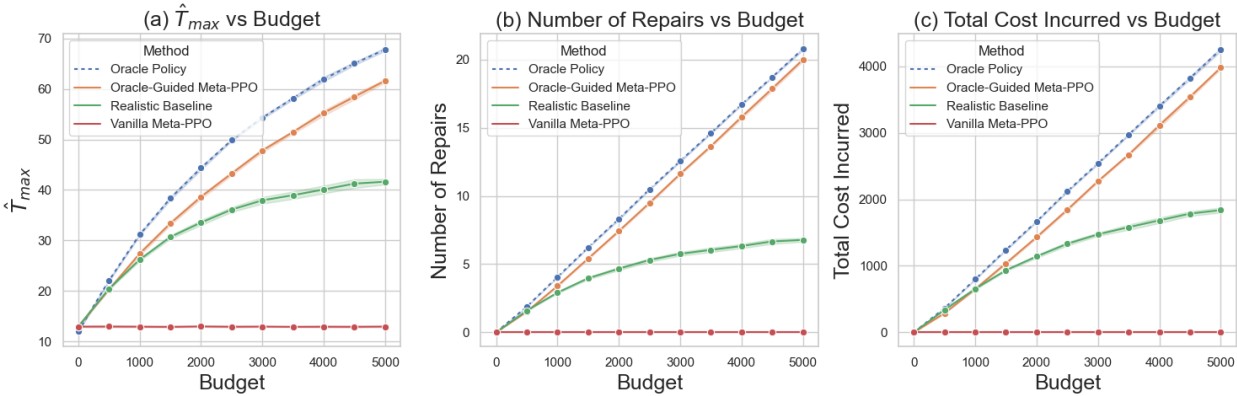

Figure 4: Performance comparison of oracle policy, oracle-guided meta-PPO, realistic baseline and vanilla meta-PPO. (a) Comparison of $\hat{T}_{max}$ values for all 1000 components across different budget values allocated to each component. (b) Comparison of average number of repairs performed by the agent under each of the four policies. (c) Comparison of average total cost incurred by the agent over the planning horizon for each of the four policies.

### 5.1.1  Analysis of Maintenance Policy

We begin by analyzing the performance of the maintenance policy derived using the proposed oracle-guided meta-PPO strategy for a single-component POMDP representing a component $i$ of the 1000 components. This policy is compared with the performance of the oracle policy on the corresponding component MDP. Since the oracle policy has full observability of the state, it is expected to always perform better than the proposed approach. Additionally, we evaluate two baseline policies:

1. A heuristic policy often used in practice (Lam & Yeh, 1994; Straub, 2004) where the agent performs inspections at regular intervals and repairs the component when its expected belief about the Condition Index (CI) falls below a predefined threshold. The realistic baseline has two hyperparameters: the inspection interval $T_{ins}$ and the repair threshold $\theta_{rep}$. We tune these hyperparameters by grid search on a held-out validation set of component instances drawn from the same distribution as the test set but not used for reporting final results. Specifically, we evaluate $T_{ins} \in \{1, \ldots, 10\}$ and $\theta_{rep} \in \{5, 10, \ldots, 50\}$ using mean survival time over validation rollouts as the selection criterion. Ties are broken in favor of less aggressive inspection. The selected values for the infrastructure experiments are $T_{ins} = 5$ and $\theta_{rep} = 15$, which are then fixed for the reported test results.

2. A vanilla meta-PPO agent that is trained on the same subset of component-budget pairs as the oracle-guided agent, but without an oracle policy.

Both the oracle-guided meta-PPO and vanilla meta-PPO are trained for 2M time steps each, with an Adam stepsize of $10^{-4}$, a minibatch size 128, policy update horizon of $T = 4096$ and discount factor 0.95. All other hyperparameters follow those used in Schulman et al. (2017). We perform 100 simulations for this component to obtain the corresponding $T_{max}^i$ values, which are then averaged over the runs for a given budget value allocated to the component. This process is repeated for all 1000 components and the run-averaged $T_{max}^i$ values are then averaged across components. We compare this average denoted by $\hat{T}_{max}$ for 11 different budget values ranging from 0 to 5000 units, along with the average number of repairs performed by the agent and the average cost incurred over the planning horizon. Figure 4 illustrates a comparison of these metrics for all four policies. We observe that the proposed approach significantly outperforms the baselines.

The oracle-guided meta-PPO agent nearly matches the performance of the oracle policy for all 3 metrics, presumably due to the low inspection costs of the components. If inspection costs were significantly higher, the agent's performance would likely diverge from the oracle policy, which is an expected outcome given the budget constraints. We also infer that the vanilla meta-PPO agent has only learnt to not violate the budget constraint by not performing any repairs. These results demonstrate the value of incorporating an oracle policy into the training of a reinforcement learning agent.

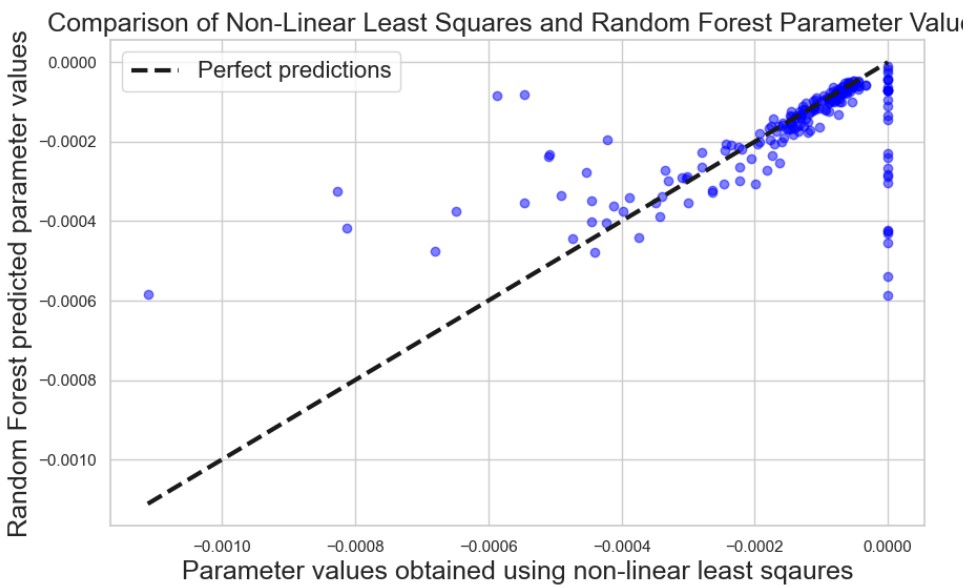

Figure 5: Performance of random forest model for predicting the value of parameter $\beta$ for a test dataset of 200 components. The horizontal axis represents parameter values obtained via non-linear least squares and vertical axis represents predicted values. The dotted line represents the $y = x$ line, i.e., perfect predictions.

### 5.1.2   Analysis of Budget Allocation

Next, we demonstrate the effectiveness of our random forest-based budget allocation strategy. We compare it with a baseline approach that allocates budgets proportional to the ratio of a component's replacement cost to its $\mathbb{E}[T]$. For a component $i$, we model its $\mathbb{E}[T_{\max}^i]$ using $\tilde{T}_{\max}^i$ as given in (12) (see Appendix A for justification of this exponential form). The parameters $\alpha^i$ and $\gamma^i$ can be estimated directly by considering the boundary conditions: $\gamma^i$ is estimated by substituting $b^i = 0$, representing the scenario where no budget is available, and $\alpha^i$ is determined by substituting $b^i = \infty$, corresponding to the scenario of unlimited budget, where the supremum of $T_{\max}^i$ ($\sup_{b^i} T_{\max}^i = H = 100$) is reached. We then train a random forest regressor to estimate parameter $\beta^i$. The training dataset is created by performing non-linear least squares regression on 11 distinct $(T_{max}^i, b^i)$ pairs each for 800 components. These pairs correspond to the run-averaged $T_{\max}^i$ values and the respective budget values $b^i$ from Section 5.1.1. The input to the random forest model is a vector consisting of the shape and scale factors of the Weibull distribution, which represent $\mathbb{E}[T]$ and $\sigma^2_{\mathbb{E}[T]}$, along with the replacement and inspection costs for a given component $i$. If a different distribution was used to model the transition probability, we would similarly extract the parameters, $\mathbb{E}[T]$ and $\sigma^2_{\mathbb{E}[T]}$, for inclusion in the input vector. Figure 5 shows the prediction performance of the random forest model for a test dataset of 200 components which were not encountered during training. We see that most points on the plot are very close to the perfect prediction line and bad predictions are few in number (29 out of 200 for error threshold of $10^{-4}$). The random forest model achieves a mean squared error (MSE) $= 1.8 \times 10^{-8}$ for this test dataset. Note that the non-linear least squares regressor constrains $\beta^i$ to be $\leq 0$ and hence for some components we observe that $\beta^i = 0$. We use this trained random forest model to estimate $\tilde{T}_{max}^i$ for all 1000 components. Finally, using these approximated expressions, we solve the constrained maximization problem described in (13) to obtain the appropriate budget allocation for the components. We quantify

Table 1: Maximum time $T_{\max}$ (steps), averaged over 100 runs, under random forest and baseline budget allocations.

| Approach | $T_{max}$ |
|---|---|
| Random Forest Budget Allocation | 22,009.5 |
| Baseline Budget Allocation | 16,445.4 |

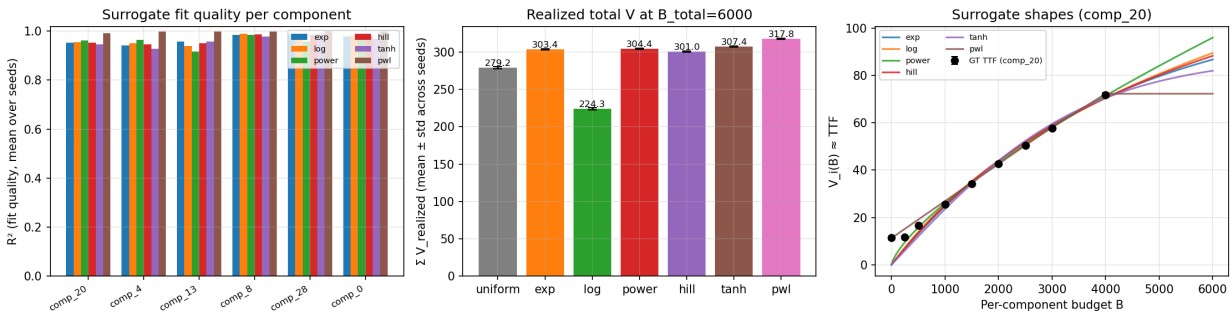

Figure 6: In-distribution surrogate-family ablation. Left: mean $R^2$ fit quality for each surrogate family across six representative components. Middle: realized total survival time obtained after solving the global allocation problem with each fitted surrogate at $B_{\text{total}} = 6000$. Right: representative fitted budget-value curves for one component. Several monotone concave saturating families achieve similar downstream performance; tanh is slightly higher among parametric families, while the exponential surrogate remains competitive and is the representative family used in the main pipeline.

the performance of the random forest budget allocation and the baseline budget allocation algorithms by calculating the $T_{max} = \sum_i T^i_{max}$ and averaging it over 100 runs. For a fair comparison, these values are obtained using the oracle-guided meta-PPO approach for both allocation schemes.

Table 1 shows the $T_{max}$ values achieved by both allocation approaches. The random forest budget allocation vastly outperforms the baseline approach. Furthermore, Figure 9 presents violin plots showing the distribution of the $T^i_{max}$ values achieved under the proposed and baseline budget allocations for all 1000 components. We observe that there are more components with higher $T^i_{max}$ values for the random forest budget allocation approach. Preliminary experiments on alternative objective formulations, such as the *maxmin* approach given by (3), also indicate that the proposed method consistently outperforms the baseline.

**Surrogate-family ablation.** We next evaluate how sensitive the allocation stage is to the surrogate used to approximate each single-component budget–value curve. For six representative infrastructure components, we fit several monotone concave surrogate families to the empirical values computed on a discrete budget grid, and then solve the same global allocation problem using each fitted family. We consider six surrogate families: exponential and hyperbolic-tangent functions as bounded saturating response curves; logarithmic and power-law functions as standard diminishing-returns curves; Hill/Michaelis–Menten functions as classical bounded response models used for saturation effects (Hill, 1910; Michaelis & Menten, 1913); and piecewise-linear concave interpolation as a non-parametric concave baseline. These families are chosen to separate two questions: whether the allocation stage needs the specific exponential form, and whether it benefits from the broader class of monotone concave saturating response curves.

Figure 6 shows the in-distribution setting, where each surrogate is fit and evaluated on the same budget range. All bounded saturating families fit the single-component curves well and yield similar downstream allocation performance. In particular, the exponential surrogate attains a realized total survival time of 303.4, which is close to power-law (304.4), Hill (301.0), and tanh (307.4), and all improve over uniform allocation (279.2). The tanh surrogate is the best parametric family in this setting, but its improvement

Figure 7: Out-of-distribution surrogate-family ablation. Surrogates are fit only on the low-budget range and then used to allocate a larger total budget. Left: mean $R^2$ fit quality on the observed range. Middle: realized total survival time after allocation at $B_{\text{total}} = 8000$. Right: representative fitted budget–value curves for one component. Exponential, tanh, and Hill/Michaelis–Menten remain stable under extrapolation, whereas the piecewise-linear surrogate degrades substantially outside the observed range.

over exponential is modest. The piecewise-linear concave surrogate performs best overall (317.8), which is expected because it directly interpolates the observed grid. By contrast, the logarithmic surrogate performs noticeably worse (224.3), indicating that its shape is less well matched to these budget–value curves.

Figure 7 shows a more demanding extrapolation setting, where surrogates are fit only on low-budget data ($B \leq 1500$) and then used to allocate a larger total budget $B_{\text{total}} = 8000$. Here the exponential surrogate attains 343.8, essentially tied with tanh (344.2) and comparable to Hill (341.6), while improving over uniform allocation (316.4). In contrast, the piecewise-linear concave surrogate drops to 228.1, reflecting the difficulty of extrapolating reliably beyond the observed budget range. Taken together, these results show that the method is not tied to the specific exponential parameterization: several monotone concave saturating families work well, while a parametric surrogate is preferable when the allocation problem requires extrapolation beyond the fitted budget range. We retain the exponential surrogate in the main pipeline because it is simple, stable under extrapolation, and already integrated with the random-forest parameter-prediction model; the ablation supports this as a representative choice from a robust class of saturating concave surrogates, rather than as the uniquely best family. Appendix B provides the corresponding numerical tables and additional details.

**Static versus dynamic budget allocation.** We also test whether redistributing the residual budget during execution materially improves performance. In this experiment, we use five representative infrastructure components with $B_{\text{total}} = 5000$ and compare the proposed static allocation against periodic reallocation every $K$ steps. Figure 8 summarizes the results. Static allocation obtains $\sum_i T_{\max}^i = 267.5 \pm 20.0$. Periodic reallocation obtains $269.7 \pm 28.1$ for $K = 5$, $275.7 \pm 21.5$ for $K = 10$, $270.9 \pm 22.7$ for $K = 25$, and $277.5 \pm 28.1$ for $K = 50$. Thus, the best reallocation setting improves mean survival time by 3.7%, but this gain is within one standard deviation of the static policy. The runtime overhead is substantially larger: over 50 episodes, reallocation costs 18.4s for $K = 50$, 182.9s for $K = 10$, and 375.8s for $K = 5$, dominated by refitting per-component surrogates from the current state. We therefore retain static allocation as the main method: it preserves the decomposition that enables scalability, while periodic reallocation provides only modest additional survival-time gains at substantially higher runtime cost. Detailed protocol and numerical values are given in Appendix C.

### 5.1.3 Analysis of Time Complexity

Finally, we analyze the time complexity of our proposed approach for varying number of components $N$. As mentioned earlier, our method comprises of four major steps:

1. **Random Forest** regression for estimating $\tilde{T}_{max}^i$ for each component $i$.

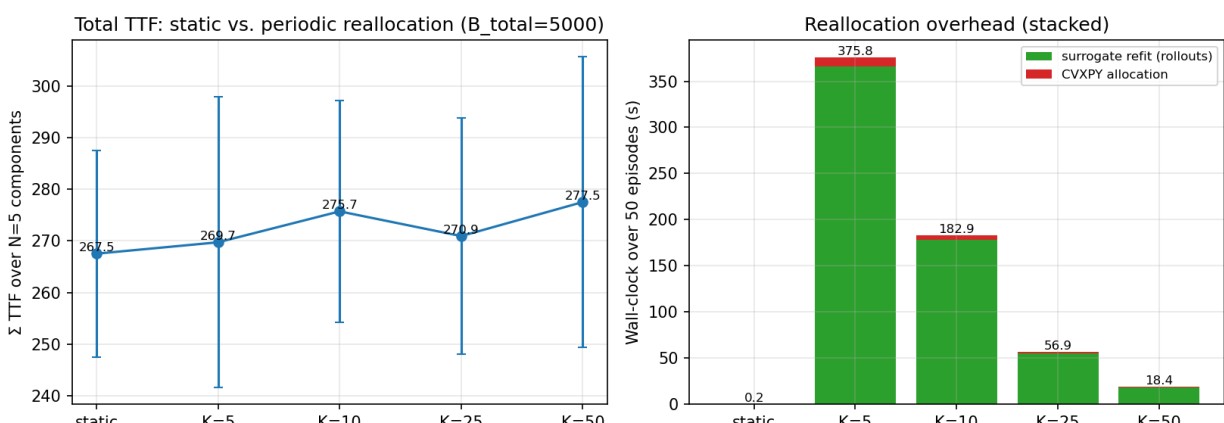

Figure 8: Static versus periodic budget reallocation. Left: total survival time over five infrastructure components. Right: reallocation overhead over 50 episodes, decomposed into surrogate-refit and CVXPY-allocation time. Periodic reallocation gives modest mean survival-time gains, but repeated surrogate refits dominate wall-clock cost.

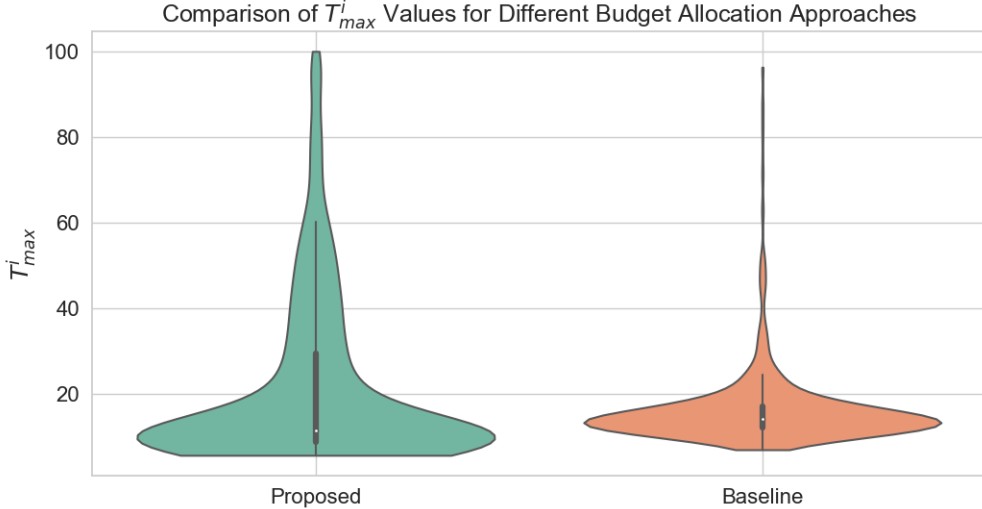

Figure 9: Performance comparison of random forest-based budget allocation and baseline budget allocation for all 1000 components for an overall budget of 500,000 units.

2. **Budget Allocation** among components via constrained optimization.

3. **MDP Value Iteration** for each component-budget pair to obtain the corresponding oracle policy.

4. **Oracle-Guided Meta-PPO** to approximately solve each component POMDP.

Table 2 presents the times taken for running each of the four processes, with different number of components. The time complexity experiments were performed in Python on a laptop running MacOS with an M2 chip @3.49GHz CPU and 8GB RAM. The times taken for random forest and budget allocation steps are negligible compared to those for performing value iteration and generating optimal policies through meta-PPO. The value iteration is applied to each component independently and hence scales linearly with the number of components. Similarly, Step 4 involves applying the pre-trained policy to each component separately and

Table 2: Time taken (in seconds) for running each process with varying numbers of components, averaged over 10 runs.

| Number of Components | Random Forest | Budget Split | Value Iteration | Meta-PPO |
|---|---|---|---|---|
| 1 | 0.9724 | 0.9046 | 113.7227 | 2.8885 |
| 2 | 0.8870 | 0.8314 | 116.3281 | 3.0858 |
| 5 | 0.8719 | 0.8207 | 135.3953 | 4.7940 |
| 10 | 0.8762 | 0.8132 | 280.4909 | 9.5495 |
| 20 | 0.9534 | 0.8997 | 451.2948 | 16.2575 |
| 50 | 0.9449 | 0.8916 | 1208.1000 | 33.7387 |
| 100 | 0.9324 | 0.9171 | 2389.5641 | 64.6809 |
| 500 | 0.9575 | 1.2226 | 10269.1037 | 313.9742 |
| 1000 | 0.9599 | 1.6232 | 20612.1734 | 627.7477 |

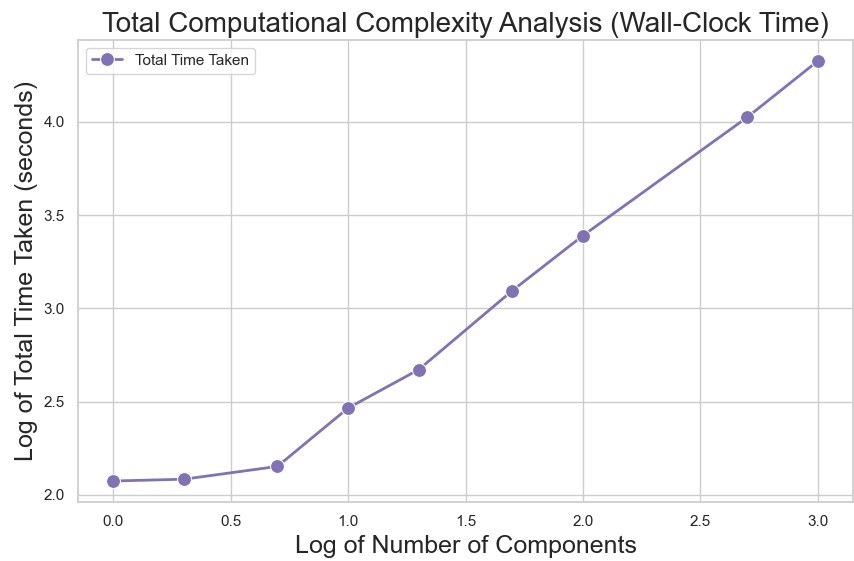

Figure 10: Log-log plot of computational complexity of the proposed approach for varying numbers of components.

thus is also linear in the number of components. Consequently, we expect that the time complexity of our algorithm is linear in the number of components, i.e., $O(n)$. This expectation is confirmed by the log-log plot of computational complexity shown in Figure 10. Our algorithm's performance is thus significantly faster as compared to existing POMDP solvers which would be exponential in the number of states and thus doubly exponential in the number of components (Silver & Veness, 2010), (Pineau et al., 2003). If the problem is approached directly as a single POMDP, it will have a prohibitively vast state space of approximately $10^{2000}$ states. Previous work by Vora et al. (2023) demonstrated that standard methods indeed become computationally intractable after a few components due to this combinatorial explosion.

### 5.1.4 Comparison with Vora et al. (2023)

Finally, we compare the proposed approach with the welfare-maximization method of Vora et al. (2023), which is the closest predecessor to our decomposition strategy. For clarity, we refer to this baseline as the *POMCP-welfare baseline* throughout this subsection. The POMCP-welfare baseline estimates a per-component budget-value curve using POMCP over a budget grid, solves a welfare allocation problem, and then executes POMCP under the allocated component budgets. This provides a strong comparison point

because it uses the same underlying idea of budget-value curves, but retains repeated POMCP computation during value-curve estimation and execution.

Table 3 reports the comparison for $N = 5$ and $N = 10$, the largest settings for which repeated POMCP-based budget-value estimation was computationally practical in our implementation. The two methods achieve nearly identical total survival times. For $N = 5$, the POMCP-welfare baseline obtains $268.1 \pm 17.2$, while the proposed method obtains $267.5 \pm 20.0$. For $N = 10$, the corresponding values are $335.1 \pm 19.2$ and $334.9 \pm 21.4$. The main difference is computational cost: the POMCP-welfare baseline requires 9565.86s for $N = 5$ and 9709.21s for $N = 10$, whereas the proposed method requires 141.88s and 291.73s, respectively. Thus, the proposed approach matches the solution quality of the POMCP-welfare baseline while reducing wall-clock time by $67.4\times$ for $N = 5$ and $33.3\times$ for $N = 10$. Detailed per-component plots and protocol details are provided in Appendix D.

Table 3: Comparison with the POMCP-welfare baseline of Vora et al. (2023). The proposed approach achieves comparable total survival time while substantially reducing wall-clock time.

| $N$ | $\sum_i T_{\max}^i$ | | Wall-clock time (s) | | Speedup |
|-----|---------------------|----------|---------------------|----------|---------|
|     | POMCP-welfare       | Proposed | POMCP-welfare       | Proposed |         |
| 5   | $268.1 \pm 17.2$    | $267.5 \pm 20.0$ | 9565.86     | 141.88   | $67.4\times$ |
| 10  | $335.1 \pm 19.2$    | $334.9 \pm 21.4$ | 9709.21     | 291.73   | $33.3\times$ |

These results clarify that the POMCP-welfare baseline is applicable and competitive at small scales, but the repeated POMCP computations make it difficult to scale to the 1000-component regime considered in our experiments. In contrast, the proposed method preserves comparable solution quality while replacing repeated online tree search with a learned component-level policy and a scalable budget-allocation pipeline.

## 5.2 Implementation and Evaluation for Financial Loss-Budget Management Scenario

Our second experimental scenario addresses a portfolio risk management task. We use daily price data for the S&P 500 constituent stocks over a two-year window, reserving the final $T = 120$ trading days for evaluation and using earlier data for training. The core component of the POMDP is an unobserved latent state defined per component as a **debit-only loss budget (health)**, $s_t \in [0, 100]$. Each component receives a small loss budget: on a day with a negative return, $s_t$ is debited proportionally and decreases; on a non-negative day, $s_t$ is unchanged; the state does not self-recover. Health increases only when the agent executes **recapitalize**. All actions draw from one shared, limited budget, and actions are taken when drift relative to the per-component no-loss floor becomes meaningful. This design is practice-inspired for two reasons. First, because we manage a large number of components, governance and our own policy favor a conservative stance: we avoid repeatedly allocating budget to components with recent serial losses, so the health is debit-only and does not auto-replenish. Second, it follows the risk-budgeting workflow described in Benham & Bebee (2024)—set a budget ex ante, allocate and monitor against a benchmark, and treat material drift as a trigger for action. To make the benchmark operational, we instantiate a per-component **no-loss floor**: losses are deviations that consume the per-component budget; gains are consistent with the floor and do not raise limits by themselves; replenishment occurs only through **recapitalize**. For training and evaluation, components are assumed independent.

**Actions and Costs:** The agent's action space $\mathcal{A} = \{\text{defer}, \text{inspect}, \text{recapitalize}\}$ manages the per-component loss budget (health). All actions draw from a shared, limited budget $B$ and follow a strict cost hierarchy $c_{\text{recapitalize}} > c_{\text{inspect}} > c_{\text{defer}}$:

- **Defer:** Continue with the current position. Incurs a low, continuous cost $c_{\text{defer}}$ each step. Health remains subject to depletion by negative returns.

- **Inspect:** Pay $c_{\text{inspect}}$ to obtain a precise observation of the hidden health $s_t$ for the selected component.

- **Recapitalize:** Pay the high cost $c_{\text{recapitalize}}$ to rebuild health by resetting $s_t$ to 100. This is the only action that increases health.

**Objective and Failure Condition.** The agent's objective is to learn a policy $\pi$ that maximizes its **survival time**. An absorbing failure state is triggered immediately if any component's health is exhausted, i.e., $s_t \leq 0$. For each day the agent survives, it receives a reward of $+1$. This setup forces the agent to learn a sophisticated policy that balances the continuous drain from defer costs and market losses against the high, discrete costs of inspection and recapitalization, in order to prolong its survival.

### 5.2.1 Analysis of Recapitalization Policy

We evaluate our approach on a **stock-level loss-budget** management task constructed from the S&P 500 universe. Starting from 500 constituents, we retain the subset with at least 80% daily-price coverage over the preceding three years, yielding 471 components. As in the infrastructure experiment, we reserve the final $T = 120$ trading days for evaluation and use earlier data for model training.

**State, actions, and costs.** Each component $j$ is modeled as a single-component monotonic POMDP with an unobserved, debit-only **loss-budget (health)** $s_t^j \in [0, 100]$. Negative returns debit $s_t^j$ proportionally; non-negative returns leave $s_t^j$ unchanged; the state does not self-recover. The action set is $\mathcal{A} = \{\text{defer}, \text{inspect}, \text{recapitalize}\}$ with a strict cost hierarchy $c_{\text{recapitalize}} > c_{\text{inspect}} > c_{\text{defer}}$. A global budget $B_{\text{tot}}$ is shared across all components. Table 4 presents the values of the various parameters used for the experiments.

Table 4: Cost and budget settings for the stock-level scenario.

| Parameter | Value |
| --- | --- |
| Total budget $B_{\text{tot}}$ | 15,000 |
| Recapitalization cost $c_{\text{RECAP}}$ | 10.0 |
| Inspection cost $c_{\text{INSP}}$ | 0.5 |
| Defer cost $c_{\text{DEF}}$ | 0.2 |
| Number of components | 471 |
| Evaluation horizon $T$ | 120 days |

**Policies compared.** We compare four policies:

1. **Oracle**: full observability of the health $s_t$; **recapitalize** whenever $s_t < 20$ (no inspection cost).

2. **Oracle-guided meta-PPO**: the agent chooses **inspect** vs. **defer**; upon **defer**, it executes the oracle's suggested restorative/default control; upon **inspect**, it pays $c_{\text{inspect}}$ to reduce belief uncertainty. The agent learns when to buy observations and when to accept uncertainty.

3. **Baseline (Heuristic)**: fixed **inspect** every 5 trading days; if the observed $s_t < 20$, take **recapitalize**; otherwise **defer**.

4. **Vanilla meta-PPO**: trained on the same component–budget pairs as the oracle-guided agent but without oracle shaping.

**Baseline hyperparameter selection.** The heuristic baseline has two hyperparameters: the inspection interval $T_{\text{ins}}$ and the recapitalization threshold $\theta_{\text{recap}}$. We select these using the same validation principle as in the infrastructure experiment, but with a chronological split to avoid using future price information. Specifically, within the training portion of the price history, we reserve the final 20% of trading days as a validation window. For each candidate pair $T_{\text{ins}} \in \{1, 5, 10, 20\}$ and $\theta_{\text{recap}} \in \{10, 20, 30, 40\}$, we generate validation rollouts by sampling initial days from this validation window such that the full horizon fits within the window, and we evaluate average survival time under the same total budget $B_{\text{tot}} = 15,000$. The same

validation start dates and random seeds are used for all hyperparameter pairs. We choose the pair that maximizes average validation survival time, breaking ties in favor of less frequent inspection and lower recapitalization thresholds to avoid selecting an unnecessarily aggressive baseline. This procedure selects $T_{\text{ins}} = 5$ and $\theta_{\text{recap}} = 20$, which are then fixed for all reported stock-level test results.

**Budget allocation.** We allocate $B_{\text{tot}}$ across the 471 **components** using the same random-forest surrogate procedure as in the maintenance experiment: for each component $i$ we fit a concave surrogate for the map $B \mapsto \mathbb{E}[T_{\text{max}}^i(B)]$ and solve a tractable concave maximization to obtain per-component budgets.

**Training details.** Both **oracle-guided meta-PPO** and **vanilla meta-PPO** are trained for $2 \times 10^6$ timesteps with Adam step size $10^{-4}$, minibatch size 128, PPO horizon $T_{\text{PPO}} = 2048$, and discount factor 0.95. For each component and policy we run 100 simulations and report the component-level average $T_{\text{max}}^i$; we then average across all 471 components to obtain $\hat{T}_{\text{max}}$.

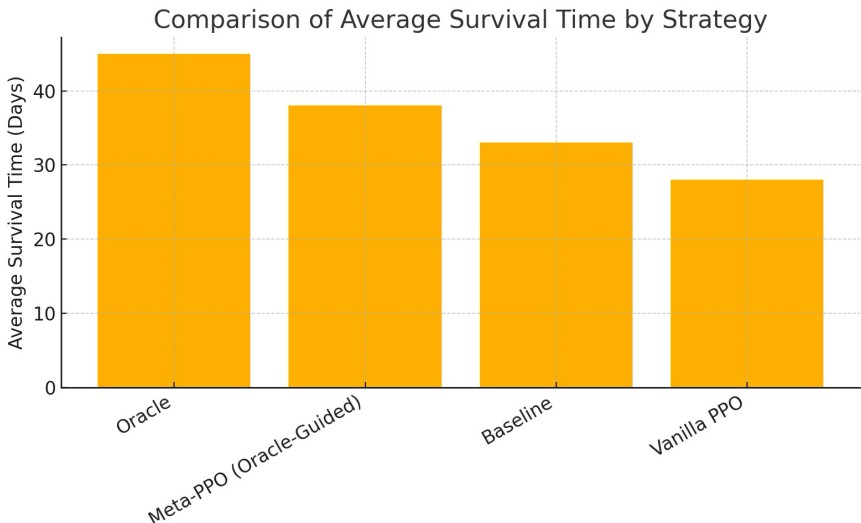

Figure 11: S&P 500 stock-level scenario: average survival time $\hat{T}_{\text{max}}$ under a shared budget $B_{\text{tot}} = 15,000$ across 471 components. Observed ordering: **Oracle > Oracle-guided meta-PPO > Baseline > Vanilla meta-PPO**.

**Findings.** Under the shared budget $B_{\text{tot}} = 15,000$ across 471 components and a 120-day evaluation horizon, we observe a consistent ordering (see Figure 11): **Oracle > Oracle-guided meta-PPO > Baseline > Vanilla meta-PPO**. The **Vanilla meta-PPO** tends to conserve budget and rarely recapitalizes, yielding the lowest survival time. The **Baseline** performs periodic inspections (every 5 trading days) and recapitalizes below the threshold but spends budget indiscriminately and misses urgent cases. By contrast, the **Oracle-guided meta-PPO** learns when to inspect versus defer and when to act, allocating budget to higher-value opportunities; it reliably outperforms the Baseline and closes a substantial portion of the gap to the Oracle upper bound.

### 5.2.2 Generalizability and Window Robustness of the Oracle-guided Meta-PPO

**Design.** We vary the number of **components** $N \in \{5, 10, 20, 100, 471\}$. For each $N$, the policy is trained on rolling 120-day train windows and evaluated both in-sample (train) and on a held-out test window. We report average survival time (days) over $r=5$ seeds; error bars denote $\pm 1$ standard deviation across seeds.

**Findings.** (1) Both train and test curves decrease as $N$ grows, reflecting budget dilution and increased problem complexity. (2) Train performance is consistently above test with a modest generalization gap that tends to widen at larger $N$. (3) Variability is non-negligible and generally larger at higher $N$.

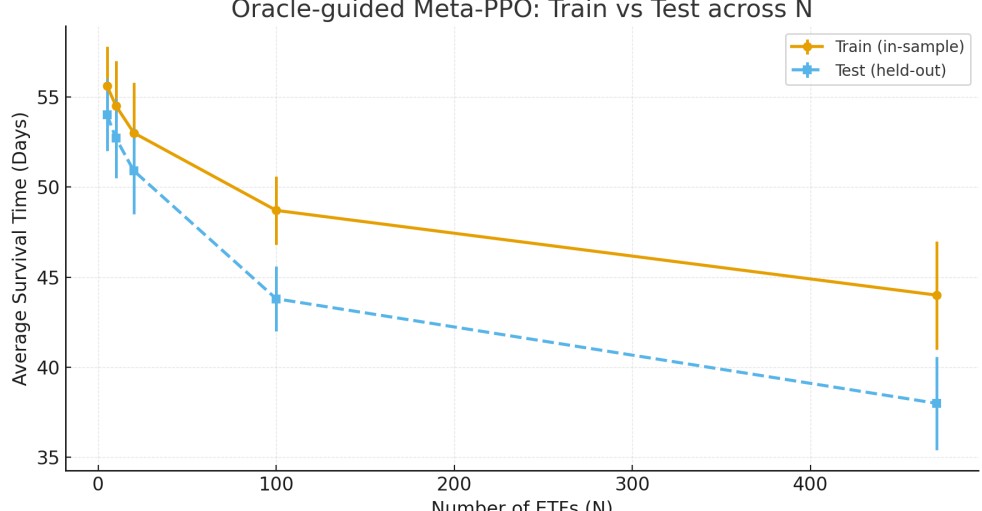

Figure 12: Oracle-guided meta-PPO: train vs. test across component set size $N$. The $y$-axis is average survival time (days); the $x$-axis is the number of components.

**Takeaway.** As can be seen from Figure 12, the oracle-guided meta-PPO exhibits window robustness: trends are consistent across train windows, and the train-to-test drop remains moderate. At small $N$, the effective exploration/interaction budget is limited, which can hinder learning; as $N$ increases, richer allocation opportunities make better use of the oracle guidance even though absolute survival time declines under a fixed total budget.

## 6   Conclusions

We proposed a scalable framework for solving *budget-constrained multi-component monotonic POMDPs*. Our chief theoretical contribution is a proof that the single-component value function is **concave in budget**, which underpins an efficient two-step solution strategy. First, a random-forest surrogate exploits that concavity to distribute the shared budget across components, thereby decomposing the large $n$-component POMDP into $n$ independent single-component POMDPs. Second, an *oracle-guided, meta-trained PPO* agent—shaped by value iteration on the fully observable counterpart—learns a near-optimal policy for each component–budget pair. Comprehensive experiments on two disparate domains confirm the framework's generality. For a 1000-component building-maintenance task, our method significantly prolongs component survival relative to baseline heuristics and approaches the performance of the oracle policy. On an ETF portfolio-rebalancing problem with draw-down–risk budgets, the same algorithm consistently preserves portfolio viability and outperforms vanilla PPO and the equal-weight baseline. Across both settings, empirical runtimes grow *linearly* with the number of components, validating the scalability predicted by our complexity analysis. Future work will focus on extending the framework's capabilities to more dynamic budget allocation schemes and more complicated hierarchical budget constraints.

### Limitations and Future Work

The proposed approach relies on two structural assumptions. First, each component follows the deterioration–restoration structure described in Section 3. Second, components are conditionally independent except through the shared budget. These assumptions are appropriate for the maintenance and portfolio settings studied here, but they exclude systems with direct physical coupling between components, such as one subsystem failure changing another subsystem's transition law. In such settings, the component-wise budget-value envelope would no longer be sufficient for global allocation, and a coupled planning or allocation model would be required. Similarly, strongly non-monotone component dynamics could weaken the concav-

ity structure used by the allocator. Extending the framework to directly coupled systems and non-monotone dynamics is an important direction for future work.

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

# A    Function Approximation of $\mathbb{E}[T_{\max}]$

We model $\mathbb{E}[T_{\max}^i]$ as an exponential function of the budget allocated to component $i$. The choice of an exponential function is motivated by its ability to capture the saturation in $\mathbb{E}[T_{\max}^i]$ values at higher budget levels, a result of the finite planning horizon $H$. Additionally, the exponential model accounts for non-zero $\mathbb{E}[T_{\max}^i]$ even when the budget is zero.

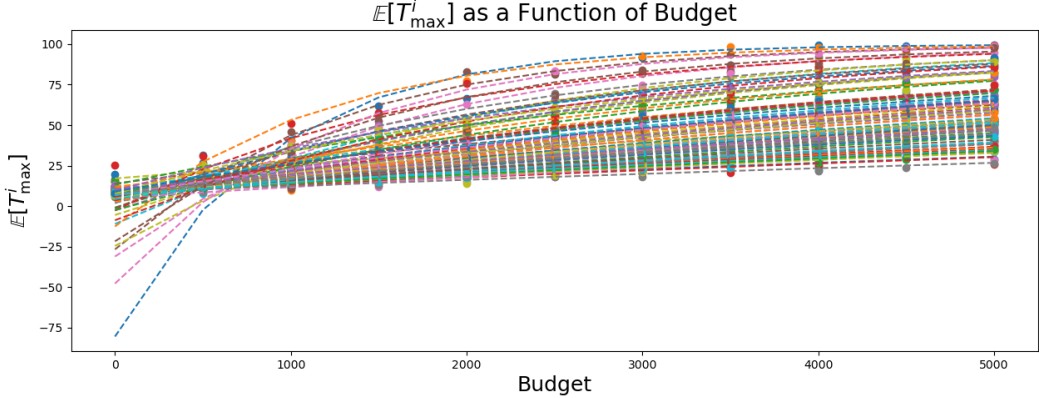

Figure 13: Exponential $\tilde{T}^i_{\max}$ curves obtained using non-linear least-squares regression.

To validate the accuracy of this exponential model for $\mathbb{E}[T^i_{\max}]$, we conducted non-linear least squares regression on 100 infrastructure components. Figure 13 illustrates the curves obtained through this regression, where $\mathbb{E}[T^i_{\max}]$ is modeled as an exponential function. The results indicate that the exponential function provides a strong approximation for $\mathbb{E}[T^i_{\max}]$, with an average coefficient of determination $R^2_{mean} = 0.899$.

## B Surrogate-Family Ablation

This appendix provides the numerical details for the surrogate-family ablation summarized in Section 5.1.2. The purpose of this ablation is to test whether the budget-allocation stage depends critically on the exponential form in (12), or whether other concave budget-value surrogates produce similar allocations.

For each component $i$, let $\mathcal{B} = \{b_1, \ldots, b_M\}$ denote the budget grid used to compute empirical single-component values, and let $\widehat{V}_i(b_m)$ denote the estimated value at budget $b_m$. For a surrogate family $g(\cdot; \theta)$, we fit parameters by solving

$$\theta^\star_i \in \arg\min_{\theta \in \Theta} \sum_{m=1}^{M} \left( g(b_m; \theta) - \widehat{V}_i(b_m) \right)^2. \tag{14}$$

The fitted surrogate $\widetilde{V}_i(B) = g(B; \theta^\star_i)$ is then used in the global allocation problem

$$\max_{B_1, \ldots, B_n \geq 0} \sum_{i=1}^{n} \widetilde{V}_i(B_i) \quad \text{s.t.} \quad \sum_{i=1}^{n} B_i \leq B_{\text{total}}. \tag{15}$$

We compare six surrogate families chosen to cover common shapes for increasing diminishing-returns curves. The exponential, hyperbolic-tangent, and Hill/Michaelis–Menten forms are bounded saturating curves, which are natural for finite-horizon value functions because $T_{\max} \leq H$. The Hill/Michaelis–Menten form is a classical saturation model originally used for cooperative binding and enzyme kinetics (Hill, 1910; Michaelis & Menten, 1913). The logarithmic and power-law forms are included as standard concave diminishing-returns baselines. Finally, the piecewise-linear concave surrogate provides a non-parametric interpolation baseline over the observed budget grid.

The surrogate families are:

$$\text{Exponential:} \quad g(B) = L(1 - e^{-kB}), \qquad L, k \geq 0, \tag{16}$$

$$\text{Logarithmic:} \quad g(B) = a \log(1 + kB), \qquad a, k \geq 0, \tag{17}$$

$$\text{Power-law:} \quad g(B) = aB^p, \qquad a \geq 0, \ 0 < p \leq 1, \tag{18}$$

$$\text{Hill/Michaelis–Menten:} \quad g(B) = L \frac{B}{K + B}, \qquad L, K \geq 0, \tag{19}$$

$$\text{Tanh:} \quad g(B) = L \tanh(kB), \qquad L, k \geq 0. \tag{20}$$

For the PWL concave surrogate, we first compute the least concave majorant of the sampled points $\{(b_m, \widehat{V}_i(b_m))\}_{m=1}^{M}$ on the fitted budget grid. Equivalently, we take the upper concave envelope of these points. The breakpoints are therefore the subset of grid points that lie on this upper concave hull, and the surrogate is obtained by linear interpolation between successive breakpoints. In the extrapolation experiment, when the optimizer queries a budget larger than the largest fitted budget point, we keep the PWL surrogate constant at the value of the final breakpoint rather than extrapolating an unsupported slope beyond the observed grid. This makes the PWL surrogate a strong interpolation baseline but a conservative extrapolation baseline.

## B.1 In-distribution evaluation

In the in-distribution setting, each surrogate is fit on the full budget grid and then used to allocate $B_{\text{total}} = 6000$. Figure 6 in the main text visualizes these results. Table 5 reports the realized total survival time after solving the allocation problem with each fitted surrogate. We list the exponential surrogate first because it is the surrogate used in the proposed method. The remaining rows show alternatives and baselines.

Table 5: In-distribution surrogate-family ablation at $B_{\text{total}} = 6000$. Each surrogate is fit on the full budget grid and then used for global allocation. The exponential row corresponds to the surrogate used in the proposed method.

| Allocation surrogate | Realized total survival time |
|---|---|
| **Exponential (proposed)** | **303.4** |
| Tanh | 307.4 |
| Power-law | 304.4 |
| Hill / Michaelis–Menten | 301.0 |
| Logarithmic | 224.3 |
| PWL concave | 317.8 |
| Uniform allocation | 279.2 |

The main conclusion from the in-distribution setting is that the proposed method is not brittle to the exact exponential parameterization. The tanh surrogate is slightly better than exponential in this particular experiment (307.4 versus 303.4), but the difference is small relative to the overall scale of the objective and does not change the qualitative conclusion: several monotone concave saturating families produce comparable allocations. We keep the exponential form in the main method because it is simple, has only two shape parameters, is stable under extrapolation, and is already the surrogate used by our random-forest parameter prediction pipeline. The PWL concave surrogate performs best in-distribution because it directly interpolates the observed budget grid; however, as shown next, this advantage does not persist when extrapolation is required.

## B.2 Extrapolation evaluation

The extrapolation setting is more demanding. Here, surrogates are fit only on the low-budget range $B \leq 1500$, but are then used to solve an allocation problem with $B_{\text{total}} = 8000$. Figure 7 in the main text visualizes these results. Table 6 reports the realized total survival time. As before, we list the exponential surrogate first because it is the surrogate used in the proposed method.

The extrapolation results explain why we use a parametric saturating surrogate in the main algorithm rather than relying directly on grid-based concave interpolation. The tanh surrogate again performs slightly better than exponential (344.2 versus 343.8), but the difference is negligible in comparison with the gap to the unstable extrapolating alternatives. Exponential, Hill, and tanh all capture the same bounded-saturation structure and therefore remain stable outside the fitted range. The PWL surrogate, which was strongest in-distribution, performs poorly because it is held constant beyond the largest observed budget point and therefore cannot represent additional marginal value outside the fitted grid. This supports the use of a simple parametric saturating surrogate in the proposed method. The exponential family is retained as the

Table 6: Out-of-distribution surrogate-family ablation at $B_{\text{total}} = 8000$, with fitting restricted to the low-budget range $B \leq 1500$. The exponential row corresponds to the surrogate used in the proposed method.

| Allocation surrogate | Realized total survival time |
|---|---|
| **Exponential (proposed)** | **343.8** |
| Tanh | 344.2 |
| Hill / Michaelis–Menten | 341.6 |
| Logarithmic | 332.3 |
| Power-law | 303.0 |
| PWL concave | 228.1 |
| Uniform allocation | 316.4 |

main surrogate because it is simple and interpretable through its asymptote and initial marginal value. The ablation should therefore be read as evidence for robustness within the class of monotone concave saturating surrogates, not as a claim that the exponential form is pointwise optimal among all possible families.

## C  Static Versus Periodic Reallocation: Additional Details

This appendix provides additional details for the static-versus-periodic budget reallocation experiment summarized in Section 5.1.2. The purpose of this experiment is to test whether the fixed a-priori budget split used by the proposed method loses substantial performance relative to a more adaptive strategy that reallocates budget during execution.

Let $K$ denote the reallocation period. At reallocation epoch $m$, occurring at time $t_m = mK$, let $B_{\text{rem}}^{(m)}$ denote the remaining total budget and let $H^{(m)} = H - t_m$ denote the remaining horizon. Periodic reallocation solves a fresh residual allocation problem

$$(B_1^{(m)}, \ldots, B_n^{(m)}) \in \arg \max_{b_1, \ldots, b_n \geq 0} \sum_{i=1}^{n} \widetilde{V}_i^{(m)}(b_i) \quad \text{s.t.} \quad \sum_{i=1}^{n} b_i \leq B_{\text{rem}}^{(m)}, \tag{21}$$

where $\widetilde{V}_i^{(m)}(\cdot)$ is a residual budget-value surrogate for component $i$, refit from the component's current state at time $t_m$ and the remaining horizon $H^{(m)}$. The resulting budget split is used for the next $K$ steps, after which the procedure repeats. The static method is the special case in which this allocation is solved only once at $t = 0$.

In our implementation, each reallocation event consists of two operations. First, for each component $i$ and each candidate residual budget value $b$ on a fixed residual-budget grid, we estimate

$$\widehat{V}_i^{(m)}(b) = \frac{1}{R} \sum_{r=1}^{R} T_{\max}^{i,(r)}(b; s_i(t_m), H^{(m)}),$$

where $T_{\max}^{i,(r)}(b; s_i(t_m), H^{(m)})$ is the survival time in rollout $r$ when starting from the current component state $s_i(t_m)$, using residual budget $b$, and planning over the remaining horizon $H^{(m)}$. We use ($R = 30$) Monte Carlo estimates to fit an exponential residual surrogate $\widetilde{V}_i^{(m)}(b)$ by the same nonlinear least-squares procedure used for the initial allocation. Second, we solve the resulting CVXPY allocation problem over the remaining total budget. The first step dominates the runtime because the residual surrogate must be recomputed from the trajectory-dependent state rather than reused from the initial state. Table 7 reports the numerical values corresponding to Figure 8 in the main text.

The best mean result is obtained by $K = 50$, corresponding to a single mid-horizon reallocation, with $277.5 \pm 28.1$ total survival time compared with $267.5 \pm 20.0$ for static allocation. This is a 3.7% improvement in the mean, but the gain is within one standard deviation of the static result. More frequent reallocation does not improve the mean further, and it substantially increases wall-clock time. These results support the

Table 7: Static versus periodic budget reallocation for $N = 5$, $B_{\text{total}} = 5000$, and 50 evaluation episodes. Runtime is total reallocation overhead over all episodes.

| Strategy | Total survival time | Reallocation overhead (s) |
|---|---|---|
| Static | $267.5 \pm 20.0$ | 0.2 |
| $K = 5$ | $269.7 \pm 28.1$ | 375.8 |
| $K = 10$ | $275.7 \pm 21.5$ | 182.9 |
| $K = 25$ | $270.9 \pm 22.7$ | 56.9 |
| $K = 50$ | $277.5 \pm 28.1$ | 18.4 |

design choice used in the main algorithm: static allocation preserves the decomposition and captures most of the observed benefit at far lower computational cost.

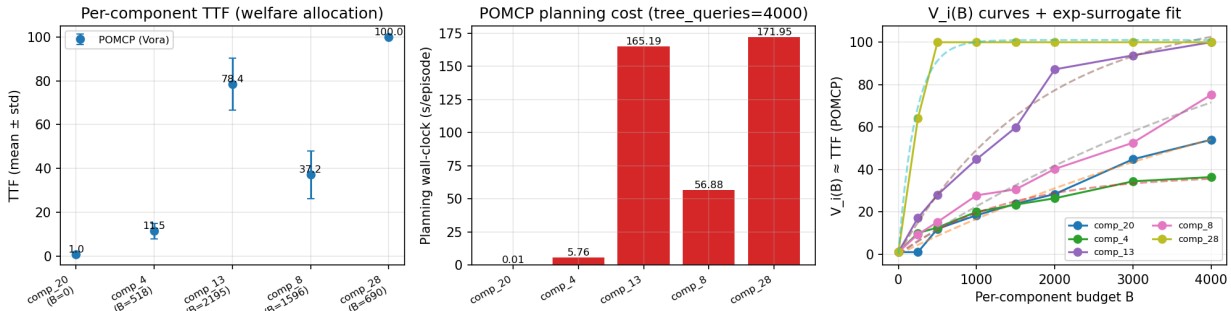

Figure 14: Detailed POMCP-welfare baseline results for $N = 5$. Left: per-component survival time under the welfare allocation. Middle: POMCP planning wall-clock cost per episode for each component. Right: estimated per-component budget-value curves and fitted exponential surrogates used by the allocation step.

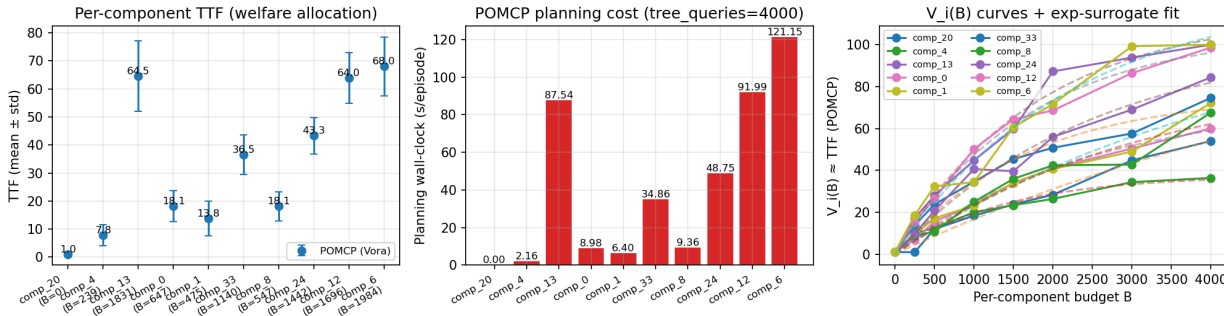

Figure 15: Detailed POMCP-welfare baseline results for $N = 10$. The same three panels are shown as in Figure 14: per-component survival time, POMCP planning cost per episode, and estimated budget-value curves with fitted surrogates.

# D  Comparison with Vora et al. (2023): Additional Details

This appendix provides additional plots for the comparison with the welfare-maximization method of Vora et al. (2023). As in Section 5.1.2, we refer to this method as the *POMCP-welfare baseline*: it estimates

per-component budget-value curves using POMCP, solves a welfare allocation problem, and then executes POMCP under the allocated component budgets. The main text reports the aggregate runtime and total-survival-time comparison. Here, Figures 14 and 15 show the per-component behavior of the POMCP-welfare baseline at $N = 5$ and $N = 10$.

Each figure has three panels. The left panel reports the per-component survival time obtained after solving the welfare allocation and executing POMCP with the allocated component budgets. The middle panel reports the POMCP planning time per episode for each component. The right panel shows the estimated budget-value curves $V_i(B)$ used by the welfare allocation, together with the fitted exponential surrogates. These plots show that the POMCP-welfare baseline is applicable and competitive at small scale, but that its cost varies substantially across components and remains tied to repeated POMCP planning.

Figure 14 illustrates the small-scale case $N = 5$. The welfare allocation assigns substantially different budgets to the five components, reflecting heterogeneity in the estimated $V_i(B)$ curves. The POMCP planning cost is also highly nonuniform across components: some components are inexpensive to plan for, while others require substantially more wall-clock time per episode.

Figure 15 shows the corresponding $N = 10$ experiment. The right panel illustrates the increasing number of budget-value curves that must be estimated as the number of components grows, while the middle panel shows that online POMCP planning costs remain significant for multiple components. Together with the aggregate runtime comparison in the main text, these plots explain the scalability bottleneck of the POMCP-welfare baseline: it can match the proposed method in solution quality at small scale, but it requires repeated tree-search computation to estimate the curves and execute the policy.

