# OpenReview forum: "Solving Truly Massive Budgeted Monotonic POMDPs with Oracle-Guided Meta-Reinforcement Learning"
_TMLR — Accepted by TMLR_

### Review · Reviewer_dytw · 2025-12-07

**Summary Of Contributions:**

The paper makes the following contributions:

- Proves, for a class of multi-component 'monotonic' POMDPs, that the value function is concave for each component.
- Leverages the concavity property to have a two-stage solver for the original POMDP. The proposed algorithm is linear on the number of components.
- Presents results comparing the proposed approach with baselines on two synthetic POMDPs and measuring runtime on increasing number of components.

**Audience:**

Yes

**Audience Explanation:**

I believe the community would find the work interesting. I only have one concern, as follows:

The concept of a monotonic POMDP (in the sense defined in this work) is interesting and potentially useful. However, to my knowledge, it is not a widely studied constraint on POMDPs. The papers cited in Section 2.3 do not appear to define this property, and the manuscript does not motivate why it merits study (beyond the second paragraph in the Introduction, which, in my view, is too general to motivate the specific definitions studied in the manuscript).

Given that this type of monotonicity in POMDPs is core to the work, the manuscript would benefit from positioning it with respect to prior work and motivating this property more. While I can imagine why the authors are interested in this, I think the manuscript should make it explicit.

**Broader Impact Concerns:**

I do not believe there are ethical implications that require adding a Broader Impact Statement.

**Claims And Evidence:**

Yes

**Claims Explanation:**

The paper is generally well-written: the assumptions and approach to deriving the method from the assumptions are explained clearly. Most of the claims are supported with clear evidence.

My concerns:
- The initial set of results (Figure 3) relies on a comparison with a “realistic baseline.” This baseline has two hyperparameters that were chosen after “extensive experiments,” and the baseline in Section 5.2.1 also has hyperparameters. Would you please comment on the methodology used to select the hyperparameters and the experiments involved?
- Furthermore, the manuscript relies on comparison with relatively primitive or classic baselines to support the claims; there is no experimental comparison with contemporary work. It appears that the work from Vora et al. (2023) is applicable, even if you expect it not to scale. Why not compare with that work?

**Requested Changes:**

1. Motivate and (if applicable) explicitly credit the work that introduced the concept of motonicity as defined in the manuscript. If this definition itself is a contribution, please say so explicitly in the manuscript.
2. In section 4, in the paragraph before section 4.1, you briefly mention an “alternate allocation strategy” that would be more expensive. What would be the benefit of this alternate approach? Please elaborate on the comparison between the approaches so that readers can better understand the tradeoffs. I believe you further comment on this in section 4.2 (“challenges scaling… our remedy is an a-priori budget distribution”), but I am left wondering if that is the tradeoff you had in mind.
3. Describe the experiments and methodology in choosing the hyperparameters of the baseline presented in, e.g., Figure 3.
4. Similarly, please describe how the hyperparameters in the baseline in Section 5.2.1 were selected.
5. It appears that the approach from Vora et al. (2023) is applicable to your problem, even if you expect it not to scale as well as your approach. Is it feasible to compare with this system?

---

> ### Author Response · Authors · 2026-05-04
> **Revision addressing monotonicity, baselines, reallocation, and comparison to prior work**
>
> Thank you for the detailed feedback. We have uploaded a revised manuscript addressing all requested changes: monotonicity positioning, the static-versus-dynamic allocation tradeoff, baseline hyperparameter selection in both experimental domains, and comparison to Vora et al. (2023).
>
> **Monotonicity motivation and positioning.** We agree that the original text under-motivated the monotonicity assumption. The revised Introduction and Sec. 2.3 now explain that monotonic POMDP refers to a deterioration-restoration structure: without intervention, condition degrades in one direction, while costly maintenance restores or partially restores it. We position this relative to structural POMDP work on ordered states/beliefs and inspection/maintenance POMDPs for deteriorating assets. We also clarify that our contribution is not the monotonicity definition itself, but the scalable treatment of many weakly coupled monotonic POMDPs sharing a global budget. A new Limitations subsection states that direct cross-component coupling or strongly non-monotone dynamics would require a different model.
>
> **Alternate allocation strategy.** We expanded the discussion before Sec. 4.1 and added an experiment in Sec. 5.1.2, with details in App. C. Online reallocation can react to realized degradation and budget consumption, but each reallocation step must solve a new joint residual allocation problem because the remaining budget is shared across components:
> $$
> \max_{b_1,\ldots,b_n\ge0}\sum_i \widetilde V_i^{(m)}(b_i)
> \quad\text{s.t.}\quad \sum_i b_i\le B_{\rm rem}^{(m)}.
> $$
> Here $\widetilde V_i^{(m)}$ is recomputed from the current state and remaining horizon.
>
> Empirically, for $N=5$, $B_{\rm total}=5000$, static allocation achieves $\sum_i T^i_{\max}=267.5\pm20.0$. Periodic reallocation gives $269.7\pm28.1$ for $K=5$, $275.7\pm21.5$ for $K=10$, $270.9\pm22.7$ for $K=25$, and $277.5\pm28.1$ for $K=50$. The best mean gain is modest, while overhead increases substantially: $18.4$s for $K=50$, $182.9$s for $K=10$, and $375.8$s for $K=5$ over 50 episodes. This supports static allocation as a deliberate scalability tradeoff.
>
> **Baseline hyperparameters.** We revised both Sec. 5.1.1 and Sec. 5.2.1 to describe baseline hyperparameter selection. In infrastructure, the realistic baseline has inspection interval $T_{\mathrm{ins}}$ and repair threshold $\theta_{\mathrm{rep}}$; we tune them by grid search on held-out component instances, evaluating $T_{\mathrm{ins}}\in\{1,\ldots,10\}$ and $\theta_{\mathrm{rep}}\in\{5,10,\ldots,50\}$, selecting $T_{\mathrm{ins}}=5$, $\theta_{\mathrm{rep}}=15$. In finance, the heuristic baseline has inspection interval $T_{\mathrm{ins}}$ and recapitalization threshold $\theta_{\mathrm{recap}}$. Within the training price history, we reserve the final 20% of trading days as validation, sample validation start dates whose full horizons fit in the window, and evaluate $T_{\mathrm{ins}}\in\{1,5,10,20\}$ and $\theta_{\mathrm{recap}}\in\{10,20,30,40\}$ under $B_{\mathrm{tot}}=15000$. This selects $T_{\mathrm{ins}}=5$, $\theta_{\mathrm{recap}}=20$, fixed for all reported tests.
>
> **Comparison with Vora et al. (2023).** We added a direct comparison in Sec. 5.1.2 and detailed plots in App. D. We refer to this baseline as POMCP-welfare: it estimates per-component budget-value curves using POMCP, solves a welfare allocation, and executes POMCP. For $N=5$, it obtains $268.1\pm17.2$, while our method obtains $267.5\pm20.0$. For $N=10$, the values are $335.1\pm19.2$ and $334.9\pm21.4$. Thus quality is comparable. The main difference is runtime: POMCP-welfare takes $9565.86$s for $N=5$ and $9709.21$s for $N=10$, compared with $141.88$s and $291.73$s for our pipeline.

---

### Review · Reviewer_WRvc · 2026-02-08

**Summary Of Contributions:**

The paper designs a method to decompose a large n-component POMDP into n independent small POMDPs, which is a very clever idea. The authors provide some theoretical results, which I found not very convincing. Then, the authors use extensive experiments to show their method outperforms baselines.

**Audience:**

Yes

**Audience Explanation:**

The idea of solving n-component POMDPs is quite novel, and I believe it will raise the interest of people who study RL.

**Claims And Evidence:**

No

**Claims Explanation:**

I think the results heavily rely on Theorem 3. However, I question the statement "which is the pointwise maximum of finitely many concave functions, and hence concave itself". For example, both x and -x are concave, but max(x,-x)=|x| is not concave.

If this theorem is not correct, the whole theoretical part should be challenged. Please let me know if I misunderstood anything.

**Requested Changes:**

From (1), please add some concrete and practical examples to justify the choice of T. It's now not well-motivated as it's hard to find real-world application scenarios.

Please explain how concavity relies on (2). For example, if we use (3), will your results need substantial changes?

The approximation in (6) lacks a theoretical guarantee or practical motivations. Also, it's better to do an ablation study on this specific form.

---

> ### Author Response · Authors · 2026-05-04
> **Revision addressing concavity proof, transition model motivation, alternate formulation, and surrogate ablation**
>
> Thank you for the careful reading. Your objection to the previous concavity proof was correct: the old proof incorrectly used the claim that the pointwise maximum of concave functions is concave. We removed that argument and substantially revised Sec. 4.1.
>
> **Corrected concavity proof.** We no longer claim concavity of the hard-budget value through the old Bellman argument. Instead, we prove concavity of the expected-cost relaxation
> $$
> V_H^{\rm soft}(b,B)=\sup_{\pi:K_H(\pi\mid b)\le B}J_H(\pi\mid b).
> $$
> The proof uses randomized mixtures of policies: if $\pi_1,\pi_2$ are feasible for budgets $B_1,B_2$, then a policy that samples $\pi_1$ with probability $\lambda$ at time 0 and $\pi_2$ otherwise is feasible for $B_\lambda=\lambda B_1+(1-\lambda)B_2$, and achieves the corresponding convex combination of returns. This establishes concavity of $V_H^{\rm soft}(b,B)$ without relying on a maximum of concave functions.
>
> We then relate this relaxation to the original hard-budget value. Since $\mathcal Y_{\rm hard}(B)\subseteq\mathcal Y_{\rm soft}(B)$, $V_H^{\rm soft}$ is an upper envelope for $V_H^{\rm hard}$. If a soft-optimal policy is hard-feasible, the values coincide. Otherwise, we bound the gap by violation probability:
> $$
> 0\le V_H^{\rm soft}(b,B)-V_H^{\rm hard}(b,B)
> \le H\,\Pr_{\pi^*_{\rm soft}}(C_H>B).
> $$
> We further show using a Doob martingale and Azuma-Hoeffding that if $K_H(\pi\mid b)\le B-\Delta$, then
> $$
> \Pr(C_H>B)\le \exp\!\left(-2\Delta^2/(Hc_{\max}^2)\right).
> $$
> Sec. 4.1 also includes an empirical validation plot showing that observed violation probabilities lie below this bound on representative infrastructure components.
>
> **Transition model $T$.** The original manuscript did not sufficiently motivate the transition structure in Eq. (1). We revised Sec. 3 to add concrete application motivation and recent references. The transition kernel abstracts a standard condition-based maintenance pattern over ordered condition states: under passive operation or inspection, condition can remain the same or deteriorate, while under maintenance/repair/replacement, the transition law changes to a repair-effect kernel that can improve condition up to $\bar{s}$. This captures infrastructure maintenance, pavement rehabilitation, structural inspection planning, and partially observable inspection/maintenance systems, where degradation is stochastic, condition is uncertain, and interventions probabilistically improve condition.
>
> **Formulation (2) versus formulation (3).** The single-component budget-value object remains useful under either objective. What changes is the global allocation. For the sum objective, we solve
> $$
> \max_{b_i\ge0}\sum_i \widetilde T^i_{\max}(b_i)
> \quad\text{s.t.}\quad \sum_i b_i\le B.
> $$
> For the max-min objective, we introduce an auxiliary variable $\eta$ and maximize $\eta$ subject to $\eta\le \widetilde T^i_{\max}(b_i)$ for all $i$. This is convex whenever each surrogate is concave, because the constraint is the hypograph of a concave function. We now point to this in the problem statement and discuss it in Sec. 4.2.
>
> **Surrogate approximation.** We added a surrogate-family ablation in Sec. 5.1.2 and App. B. In-distribution, exponential, Hill, tanh, and power-law surrogates produce similar allocation quality. Out-of-distribution, bounded parametric saturating families remain reliable, while PWL interpolation performs poorly because it cannot extrapolate beyond the observed grid. Tanh is slightly higher than exponential in these ablations, but the difference is small; we keep exponential because it is simple, stable, and compatible with the random-forest parameter-prediction pipeline. The conclusion is that the method is robust within the class of monotone concave saturating surrogates, not that exponential is uniquely optimal.

---

### Review · Reviewer_6Egb · 2026-04-22

**Summary Of Contributions:**

This paper introduces a scalable framework for solving large-scale budget-constrained monotonic POMDPs, addressing a problem that is typically computationally intractable. Specifically, the authors leverage the budget-concavity of the value functio to decompose the multi-component problem into tractable single-component subproblems. The resulting two-stage approach combines random forest-based budget allocation with an oracle-guided meta-PPO algorithm, and is validated on infrastructure maintenance and financial portfolio management tasks.

**Audience:**

Yes

**Audience Explanation:**

The paper addresses the problem of large-scale budget-constrained monotonic POMDPs that is typically computationally intractable. Research in this specific area may benefit from the findings.

**Claims And Evidence:**

Yes

**Claims Explanation:**

The algorithm has theoretical guarantees: the analysis of budget-concavity provides a solid foundation for the proposed decomposition strategy. Besides, the experiments on infrastructure maintenance and financial portfolio management tasks validate the effectiveness of the proposed method.

**Requested Changes:**

1. Provide deeper analysis or ablations on how the surrogate model (random forest) and fixed budget allocation affect optimality, especially under varying conditions.
2. Budget allocation is performed a priori rather than dynamically, potentially reducing responsiveness during execution. Can the authors explore or discuss extensions where budget allocation can adapt over time rather than being fixed a priori?
3. The method relies on monotonicity and independence assumptions, which may not hold in more complex real-world systems. Can the authors add discussions or provide experiments on less structured environments?

---

> ### Author Response · Authors · 2026-05-04
> **Revision addressing surrogate choice, dynamic allocation, and modeling assumptions**
>
> Thank you for the constructive review. We have uploaded a revised manuscript addressing the surrogate/fixed-allocation analysis, adaptive reallocation, and the scope of the monotonicity/independence assumptions.
>
> **Surrogate model and fixed allocation.** The exponential surrogate is not a structural requirement of the framework. The allocation stage only requires a monotone concave approximation of each component’s budget-value curve. We now state this in Sec. 4.2 and add a surrogate-family ablation in Sec. 5.1.2, with tables/details in App. B. We compare exponential, log, power-law, Hill/Michaelis-Menten, tanh, and piecewise-linear concave surrogates.
>
> In-distribution, where surrogates are fit on the full budget grid and evaluated at $B_{\rm total}=6000$, several concave saturating families perform similarly: exponential gives total survival time $303.4$, power-law $304.4$, Hill $301.0$, tanh $307.4$, compared with $317.8$ for the PWL concave hull and $279.2$ for uniform allocation. Out-of-distribution, where surrogates are fit only on $B\le1500$ but used at $B_{\rm total}=8000$, bounded parametric families remain stable: exponential $343.8$, Hill $341.6$, tanh $344.2$. Although tanh is slightly higher in these two ablations, the differences from exponential are small. We retain exponential because it is simple, stable under extrapolation, and already integrated with the random-forest parameter-prediction pipeline. Thus, the ablation shows that the method is not brittle to the exponential form; rather, it benefits from the broader class of monotone concave saturating surrogates.
>
> **Static versus adaptive allocation.** We agree that fixed a-priori allocation trades responsiveness for scalability. We now acknowledge this before Sec. 4.1 and add a controlled experiment in Sec. 5.1.2, with protocol details in App. C. In the adaptive variant, every $K$ steps we solve the residual allocation problem
> $$
> \max_{b_1,\ldots,b_n\ge0}\sum_i \widetilde V_i^{(m)}(b_i)
> \quad\text{s.t.}\quad \sum_i b_i\le B_{\rm rem}^{(m)},
> $$
> where $\widetilde V_i^{(m)}$ is refit from the current component state and residual horizon.
>
> For $N=5$, $B_{\rm total}=5000$, static allocation gives $\sum_i T^i_{\max}=267.5\pm20.0$. Periodic reallocation gives $269.7\pm28.1$ for $K=5$, $275.7\pm21.5$ for $K=10$, $270.9\pm22.7$ for $K=25$, and $277.5\pm28.1$ for $K=50$. The best mean gain is $3.7\%$, within one standard deviation. Runtime overhead is much larger: $18.4$s for $K=50$, $182.9$s for $K=10$, and $375.8$s for $K=5$ over 50 episodes, dominated by surrogate refits. This supports fixed allocation as the main method: it preserves the decomposition and captures most of the benefit at much lower runtime.
>
> **Monotonicity and independence assumptions.** We strengthened the Introduction, Sec. 2.3, and Sec. 3 to explain that monotonic POMDP denotes a deterioration-restoration structure common in maintenance/asset-management models. We also added a Limitations subsection clarifying that direct cross-component coupling or strongly non-monotone dynamics would require a different coupled model.

---

### Decision · Action_Editor_5FET · 2026-06-02

**Recommendation:** Accept with minor revision

**Additional Comments:**

The authors have made significant improvements to the manuscript, particularly in clarifying the theoretical underpinnings and strengthening the experimental validation. However, the presentation of the problem setup remains opaque and requires further refinement to improve readability.

Specifically, the setup needs to more explicitly bridge the gap between abstract formalism and real-world application scenarios. Furthermore, there is a recurring issue with undefined or extraneous notation; for example, the variables $d_i, q_i,$ and $m_i$ are introduced in the setup but appear to go unused in subsequent derivations. Clarifying these definitions or removing unused symbols is essential to ensure that the material is accessible and rigorous for the intended audience.

**Audience:**

Yes

**Audience Explanation:**

The findings of this paper will be of significant interest to researchers and practitioners working on budget-constrained decision-making, large-scale infrastructure maintenance, and financial portfolio management, as the proposed framework offers a scalable, computationally tractable approach to solving complex, monotonic POMDPs.

**Claims And Evidence:**

Yes

**Claims Explanation:**

The mathematical derivations have been rigorously verified, and the experimental results provide compelling evidence of the proposal's effectiveness.